# Mechanism for Vipp1 spiral formation, ring biogenesis, and membrane repair

Souvik Naskar [1,5], Andrea Merino[2,5], Javier Espadas [3], Jayanti Singh[1], Aurelien Roux [3], Adai Colom [2,4] ✉ & Harry H. Low [1] ✉

The ESCRT-III-like protein Vipp1 couples filament polymerization with membrane remodeling. It assembles planar sheets as well as 3D rings and helical polymers, all implicated in mitigating plastid-associated membrane stress. The architecture of Vipp1 planar sheets and helical polymers remains unknown, as do the geometric changes required to transition between polymeric forms. Here we show how cyanobacterial Vipp1 assembles into morphologically-related sheets and spirals on membranes in vitro. The spirals converge to form a central ring similar to those described in membrane budding. Cryo-EM structures of helical filaments reveal a close geometric relationship between Vipp1 helical and planar lattices. Moreover, the helical structures reveal how filaments twist—a process required for Vipp1, and likely other ESCRT-III filaments, to transition between planar and 3D architectures. Overall, our results provide a molecular model for Vipp1 ring biogenesis and a mechanism for Vipp1 membrane stabilization and repair, with implications for other ESCRT-III systems.

Endosomal sorting complex required for transport-III (ESCRT-III) family members are ancient membrane remodeling devices with an evolutionary lineage that traces back to the last universal common ancestor of cells[1]. Over time, the family has radiated across the tree of life, acquiring often essential and conserved functions. In eukaryotes and archaea, ESCRT-III systems drive membrane abscission during cell division[2], promote viral replication and budding[3,4] and mediate extracellular vesicle biogenesis[5,6]. Other eukaryotic functions include multivesicular body biogenesis[7] and membrane repair[8]. In bacteria, in which PspA and its paralogue vesicle-inducing protein in plastids 1 (Vipp1/IM30) were discovered as ESCRT-III homologs[1,9], these proteins function in membrane stress response and repair. PspA activity is triggered by agents that threaten inner membrane integrity, including phage, mislocalized secretins, and antibiotics[10–14], whereas Vipp1 is a plastid component in cyanobacteria, algae, and plants, in which it functions in thylakoid membrane biogenesis and repair[15–24].

ESCRT-III family members have a conserved fold consisting of five helices, α1–α5 (refs. 1,25,26). Whereas helices α1 and α2

form a characteristic hairpin motif, in some systems helices α3–α5 switch between open, intermediate, and closed conformations[27]. Some ESCRT-III family members, such as Vipp1, Vps2 (CHMP2), Vps24 (CHMP3), and Snf7 (CHMP4) supplement this fold with a membrane-binding amino-terminal motif or amphipathic helix (helix α0)[13,28–30]. Carboxy-terminal to helix α5 are less conserved elements[1] that mediate protein interactions in most eukaryotic ESCRT-III systems[25]. In this region, Vipp1 has a ~40-amino-acid C-terminal domain (CTD) that is flexible and may incorporate helix α6 (ref. 31). The CTD tunes Vipp1 polymerization dynamics both in vivo and in vitro[1,31–33]. Using the core fold as a building block, ESCRT-III family members assemble filaments where the hairpin motif of neighboring subunits stack side by side, with helix α5 binding in a domain swap across the hairpin tip[1,7,9,34,35]. This filament is used to build different supramolecular structures, including spirals[34,36–45], helical filaments[9,34,44,46], and dome-shaped rings[1,35]. In bacteria, although *Synechocystis* Vipp1 (ref. 47) and PspA[9] form planar patches, spiral filaments have not been reported, which

[1]Department of Infectious Disease, Imperial College, London, UK. [2]Biofisika Institute (CSIC, UPV/EHU) and Department of Biochemistry and Molecular Biology, University of the Basque Country, Leioa, Spain. [3]Biochemistry Department, University of Geneva, Geneva, Switzerland. [4]Ikerbasque, Basque Foundation for Science, Bilbao, Spain. [5]These authors contributed equally: Souvik Naskar, Andrea Merino. ✉e-mail: adai.colom@ehu.eus; h.low@imperial.ac.uk

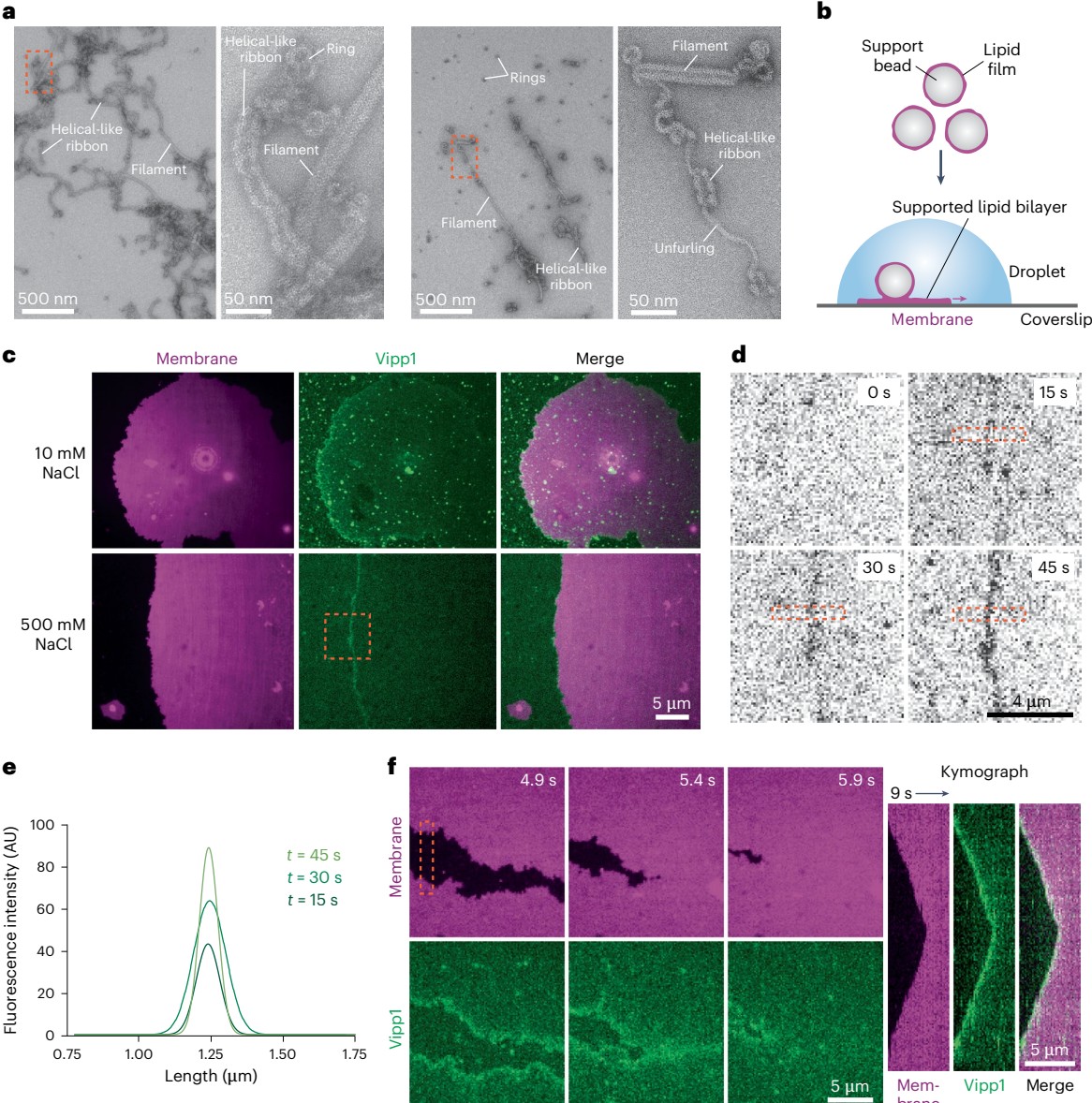

**Fig. 1 | Vipp1 is a membrane sensor recruited to highly curved and perturbed membranes. a**, Left pair, NS EM images showing Vipp1 purified in low-salt (10 mM NaCl) buffer. The area in the red dashed box is enlarged in the second image. Right pair, Vipp1 in a 500 mM NaCl buffer. The area in the red dashed box is enlarged in the second image. The unfurling of the helical-like ribbon is shown. Experiments were repeated independently more than three times. **b**, Preparation of SLBs. **c**, Fluorescent microscopy showing Vipp1$_{Alexa488}$ recruitment to the highly curved membrane edge. Vipp1$_{Alexa488}$ recruitment to the membrane edge is unaffected by ionic strength. The experiment was repeated independently three times. The area in the dashed box is shown in **d**. **d**, Timecourse showcasing dynamic Vipp1$_{Alexa488}$ recruitment to the membrane edge. **e**, Fitted curves of the fluorescence plot profile show increasing Vipp1$_{Alexa488}$ recruitment to the membrane edge. Measurements were collected from the area in the dashed box in **d**. **f**, Timecourse showing fusion of neighboring SLBs with Vipp1$_{Alexa488}$ lost from the membrane merge point. The kymograph (right) is related to the region enclosed by the dashed box. The experiment was repeated independently three times.

currently represents a key differentiating factor from their eukaryotic counterparts.

The assembly of ESCRT-III filaments is fundamental to their membrane-remodeling mechanism, with the formation of planar spirals on the membrane being a key step. Current models describe spirals as loaded springs with elastic stress accumulating owing to a preferred radius of curvature. Stress is highest at the spiral perimeter and center, where the filament is under- or over-curved, respectively. This stress, which constitutes an energy store, is theoretically sufficient to bend the membrane[41,48]. Energy minimization, through the buckling of planar spiral filaments to three-dimensional (3D) polymers such as conical spirals or helices, is directly coupled to the mechanical shaping of the bound membrane. An important component of the model depends on the sequential tilting of polymer orientation relative to the membrane[49,50], with exposure to tilted membrane-binding interfaces in the filament being a driver of membrane deformation[50,51]. Notably, by simply switching the position of the membrane-binding interface to be oriented towards the inside or outside of the tilted filament, the direction of membrane budding can theoretically be reversed with the filament binding and exerting force from the outside or inside of the membrane, respectively[49]. How ESCRT-III subunits are arranged in these planar or twisted filaments when bound to membrane remains unclear.

The dome-shaped ring structure of Vipp1, along with the observation that the rings are sufficient to bud membrane, suggests that there is an alternative mechanism by which ESCRT-III family members remodel membranes[1]. Vipp1 rings recruited to the surface of both monolayers

and precurved lipid bilayers bud the membrane by internalizing it in the ring lumen. This budding is mediated by membrane-binding domains (helix α0) lining the inner lumen so that membrane is drawn in via a capillary action-like mechanism. Although Vipp1 and PspA have the capacity to bind and hydrolyze nucleotides[35,52], this mode of ring-mediated membrane remodeling occurs passively without chemical energy turnover, at least in vitro. The capacity of Vipp1 rings to bud membrane independently of spiral springs hints that a similar process could be an unrecognized contributory factor in eukaryotic ESCRT-III membrane-remodeling processes wherein rings are assembled, albeit transiently, in the center of spirals[53].

The biogenesis pathway for Vipp1 rings is unknown. It is also unclear how Vipp1 rings, as well as other polymers such as helical filaments, might relate structurally and functionally to membrane-bound planar forms that stabilize and reduce proton permeability in liposomes[47]. These planar forms could play a fundamental part in how Vipp1 and PspA contribute to membrane stabilization and repair across bacteria. Here we use a combination of light microscopy, fast-atomic force microscopy (F-AFM), and electron microscopy (EM) to show how Vipp1 rings originate from a planar spiral progenitor. By assembling and comparing structural models for different types of Vipp1 polymer, including planar sheets, spiral filaments, helices, and rings, we suggest a model for Vipp1 ring biogenesis that has implications for how other ESCRT-III systems facilitate membrane budding.

## Results

### Vipp1 is a membrane sensor recruited to highly curved and perturbed membranes

Vipp1 from *Nostoc punctiforme* was purified in a low-salt (10 mM NaCl) buffer (Fig. 1a). This yielded various 3D polymers, including helical filaments, helical-like ribbons, and some dome-shaped rings[1]. To study Vipp1 dynamics on membrane, Vipp1 was labeled with Alexa Fluor 488 (termed Vipp1$_{Alexa448}$), introduced into a flow chamber incorporating supported lipid bilayers (SLBs; Fig. 1b), and imaged by confocal microscopy. This methodology has previously been used to characterize how the ESCRT-III protein Snf7 interacts with model membranes[41]. Whereas Snf7 formed evenly distributed patches on the membrane surface, Vipp1$_{Alexa448}$ formed a localized coating at the SLB edge (Fig. 1c). At the edge, the membrane was highly curved, with regions of poorly packed or perturbed lipid expected. Using a buffer with 500 mM NaCl promoted partial disassembly of the polymers (Fig. 1a) and reduced larger soluble fluorescent foci and background fluorescence. However, Vipp1$_{Alexa448}$ still targeted the highly curved SLB edge (Fig. 1c–e). Where neighboring SLBs fused, bound Vipp1$_{Alexa448}$ rapidly disassembled from the membrane merge point, supporting the notion of a highly dynamic interaction with the membrane edge (Fig. 1f). In summary, these data revealed that Vipp1$_{Alexa488}$ has a strong preference for highly curved and perturbed membrane, and were consistent with Vipp1 functioning as a membrane-curvature sensor.

### Vipp1 assembles dynamic networks of spirals, rings, and sheets on membrane

To further resolve the dynamics and architecture of Vipp1 on the membrane edge, we used F-AFM, which offers high temporal and spatial resolution. Consistent with the fluorescent microscopy data (Fig. 1c–f), Vipp1 accumulated at the edge of SLB patches, indicative of a sensing capability for highly curved or perturbed membrane (Fig. 2a). However, we did not observe the expected binding of 3D polymers, such as helical filaments (Fig. 1a). Instead, Vipp1 grew as dynamic planar filaments that curled anticlockwise to form spirals or sometimes rings (Fig. 2b, Extended Data Fig. 1a, and Supplementary Videos 1 and 2). The sample was originally gel-filtrated, and the filaments grew from small oligomers (or monomers) that originated from disassembled 3D polymers. Filaments grew into unpopulated membrane regions such that networks of planar sheets, spiral filaments, and rings covered the

entire lipid surface within minutes (Fig. 2c). Sometimes, the filaments split, indicative of a protofilament substructure, or nascent filaments grew alongside established filaments to form planar sheets (Extended Data Figs. 1b,c). Conversely, sheets separated, branching into filaments that curled into spirals (Fig. 2d and Extended Data Fig. 1c). The membrane height offset for both sheet and spiral filaments was ~5.5 nm, suggesting a close structural relationship (Fig. 2e). Filaments grew at a mean rate of 24 ± 19.6 nm s$^{-1}$ (mean ± s.d.; $n$ = 124 independent measurements; Fig. 2f) and with a generally uniform width, with a mean of 13.4 ± 0.9 nm ($n$ = 13; Fig. 2g). The spiral mean diameter was 82.7 nm ± 37.8 ($n$ = 278; Fig. 2h), with a mean area of 6,485 ± 6,303 nm$^2$ ($n$ = 278; Fig. 2i). These spirals were reminiscent of Snf7 spirals[41], although when packed they did not deform into polygons like Snf7, indicating that the Vipp1 filament is stiffer. They also did not fragment towards the spiral perimeter, which was characteristic of Snf7. The spirals curled inward in either Archimedes or exponential forms until the filaments reached a curvature limit, beyond which further curling was impeded, resulting in filament merging and formation of closed rings (Figs. 2d and 3a). Spiral maturation and ring biogenesis correlated with increasing height offset between spiral inner turns and the membrane (Fig. 3b, Extended Data Fig. 1a, and Supplementary Video 3). In mature spirals, rings protruded 1.0 ± 0.2 nm ($n$ = 16) above the surrounding spiral filaments (Fig. 3c) and eventually detached from the parent spiral (Fig. 3a,d and Extended Data Fig. 1a). These rings, which originated from spirals in low-salt buffer conditions (10 mM NaCl) and had a mean diameter of 37.0 ± 3.9 nm ($n$ = 39), were termed Vipp1 rings$_{LS}$ (Fig. 3e,f). Periodically, rings formed spontaneously on the curved edge of SLBs or on isolated lipid micro-patches in the absence of a parent spiral (Extended Data Fig. 1a,d,e). Here, the nascent Vipp1 filament grew spatially constrained by the membrane support or surrounding planar filaments. In these instances, the mean ring height was 6.0 ± 0.6 nm ($n$ = 22), which was similar to Vipp1 rings$_{LS}$ (Extended Data Fig. 1e). However, the mean diameter was generally wider (49.1 ± 7.8 nm; $n$ = 39) than that of Vipp1 rings$_{LS}$ (Fig. 3e,f), showing how maximum bending curvature was not generally achieved without the corralling effect of a spiral.

### Vipp1 rings scan and bind damaged membrane

Dome-shaped rings have been previously characterized, revealing symmetries ranging from $C_{11}$ to $C_{17}$ with diameters from 24 nm to 34 nm, respectively[1]. Larger rings with $C_{20}$ symmetry and a diameter of 41 nm, were also prevalent in the sample, with the largest ring class being ~43 nm diameter (Extended Data Fig. 2a). To obtain these pre-assembled rings, the sample was purified in a high-salt buffer (50 mM NaCl)[1]. These rings were termed Vipp1 rings$_{HS}$ to distinguish them from Vipp1 rings$_{LS}$. Given potential similarities in architecture between Vipp1 rings$_{HS}$ and Vipp1 rings$_{LS}$, we characterized Vipp1 rings$_{HS}$ using F-AFM. Vipp1 rings$_{HS}$ exposed to SLBs bound the membrane surface and maintained their pre-assembled ring structure (Fig. 3g and Extended Data Fig. 2b). Intriguingly, Vipp1 rings$_{HS}$ targeted the highly curved edge of membrane patches or ruptures in the lipid. Vipp1 rings$_{HS}$ therefore have the remarkable capability of scanning membrane surfaces for damaged regions before targeting them for binding. As determined using F-AFM, Vipp1 rings$_{HS}$ had a mean diameter of 35.5 ± 2.9 nm ($n$ = 20; Fig. 3e,f), which correlated with the mean diameter of Vipp1 rings$_{LS}$ and the larger ring diameters observed by EM (Extended Data Fig. 2a)[1]. Vipp1 rings$_{HS}$ had a mean height of 9.6 ± 2.2 nm ($n$ = 46; Extended Data Fig. 2c), which was ~3 nm higher than Vipp1 rings$_{LS}$, but lower than Vipp1 rings$_{HS}$, which were between ~15 and 21 nm for $C_{11}$–$C_{17}$ symmetries as determined by cryo-EM[1]. For the latter, the difference was likely due to disassembly of the lower rungs of the dome-shaped rings into the membrane once bound, although compression from the F-AFM tip cannot be ruled out. Overall, Vipp1 rings$_{HS}$ function as membrane-sensing scaffolds and are broadly similar in shape and form to Vipp1 rings$_{LS}$ when measured by F-AFM (Fig. 3h).

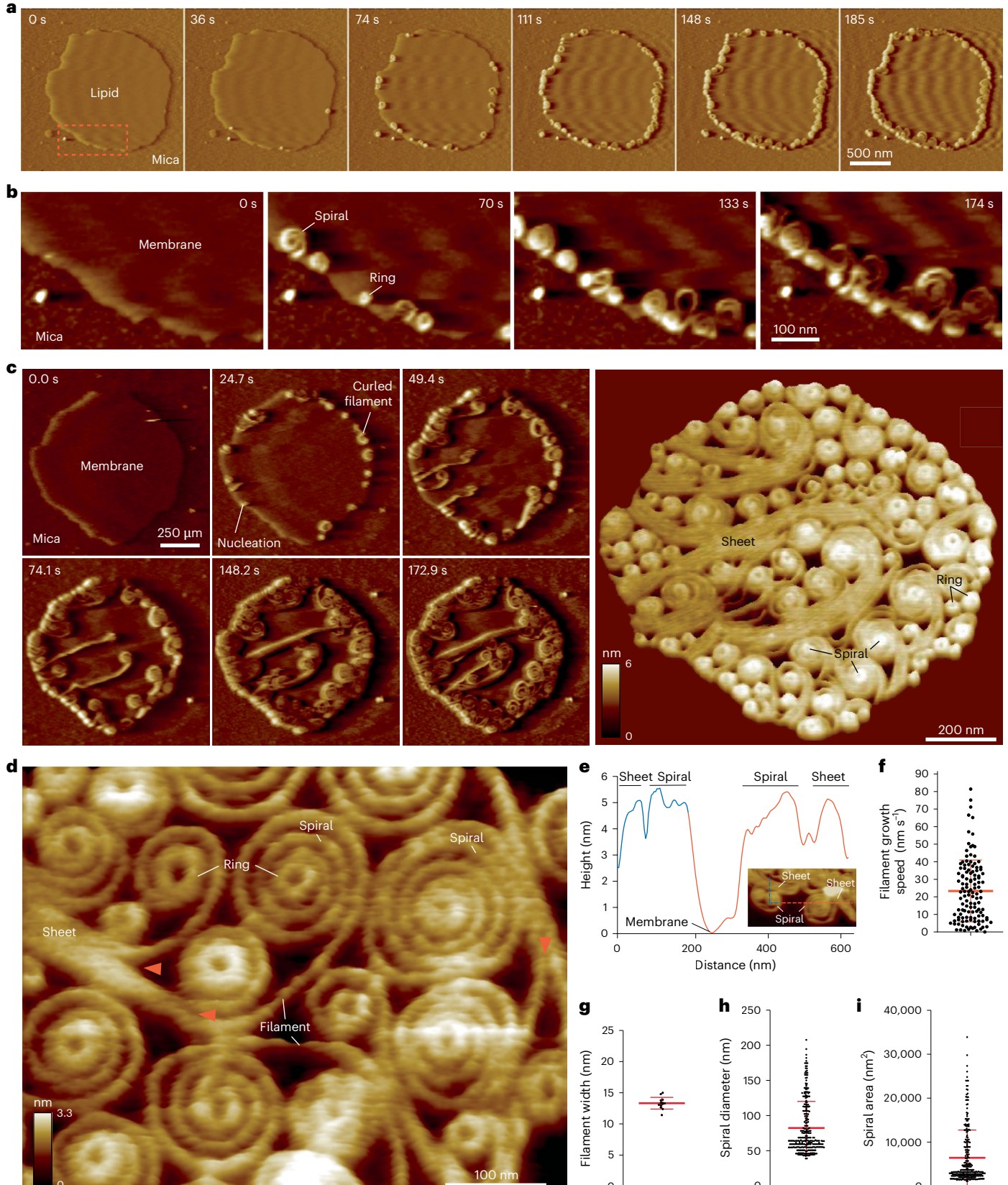

**Fig. 2 | Vipp1 assembles dynamic networks of spirals, rings and sheets on membrane. a**, F-AFM phase timecourse showing Vipp1 recruitment to the highly curved edge of membrane patches. Scan rate, 70 Hz; 256 × 256 pixels. The area in the dashed box is enlarged in **b**. **b**, Spiral and ring formation localized to the membrane edge. Scan rate, 70 Hz; 256 × 256 pixels. **c**, Left, phase timecourse showcasing a dense network of sheets, spirals, and rings that ultimately cover the entire membrane plane. Right, average of six F-AFM height images.

Scan rate, 120 Hz; 256 × 256 pixels. **d**, Average F-AFM height image showing Vipp1 sheet, spiral, and ring detail. Red arrows mark the sheet branching into filaments ~13 nm wide. Scan rate, 20 Hz; 256 × 256 pixels. **e**, Vipp1 sheet and spiral filament height offset from the membrane. **f–i**, Quantification of Vipp1 filament and spiral characteristics. $n = 124, 13, 278,$ and 278 independent measurements for panels **f**, **g**, **h**, and **i**, respectively. Error bars show one s.d. of the mean.

Next, we examined the effect of low salt concentrations (10 mM NaCl) on Vipp1 rings$_{HS}$. Intriguingly, Vipp1 rings$_{HS}$ were not observed binding the lipid. Instead, they disassembled and nucleated spirals, sheets, and rings that grew from the membrane edge (Extended Data Fig. 2d,e). Negative-stain EM (NS EM) verified the polymeric state of Vipp1 rings$_{HS}$ in the low-salt buffer (Extended Data Fig. 2f), although rings had an increased tendency to clump or unfurl into quasi-helical ribbons, showcasing the effect of low salt on destabilizing the rings and promoting a transition towards helical-like filaments.

## Vipp1 spiral filaments and planar sheets have closely related lattices

To analyze the relationship between Vipp1 sheet and spiral filament ultrastructure under low-salt conditions (10 mM NaCl), we used lower-velocity AFM, which revealed remarkable detail. Sheets were organized into parallel stripes or ridges, spaced 54 Å apart (Fig. 4a). Orthogonal to the ridge lines, parallel seams spaced 122 Å apart could be discerned in the sheets, which was consistent with the width of the filaments in spiral turns (Fig. 2g). This supports a model in which sheets form planar crystalline superstructures comprising merged filaments that were ~12–13 nm across and are predisposed to branch and spiral at the seam points. Analysis of spiral filaments also revealed parallel ridges spaced 54 Å apart (Fig. 4b). Neighboring spiral turns were sometimes merged, indicating that filaments might transition into localized sheet patches under different conditions. To further resolve sheet ultrastructure, Vipp1 was exposed to lipid monolayers. In contrast to the SLB system where Vipp1 3D polymers did not bind directly to the membrane (apart from Vipp1 rings$_{HS}$), 3D polymers such as rod-like filaments were sometimes observed bound to the lipid monolayer surface (Fig. 4c). These presumably originated as ordered helical filaments in the initial sample (Fig. 1a). They often had tips that appeared to be flattening and merging with the surrounding milieu. Remarkably, close inspection of the background monolayer revealed striped filaments forming spirals or concentric turns that merged into wider sheets. Rings were usually observed in the spiral center. After an incubation of ~15 min, this mosaic of filaments spanned many micrometers in diameter (Fig. 4d). By extracting and aligning filament subsections of 35 nm$^2$, class averages were generated, revealing a 54 Å spacing between filament stripes. As the spacing and the width of the filaments was ~12–14 nm, we concluded that these Vipp1 filaments were equivalent to those formed on SLBs. Overall, these data support a model wherein spiral filaments and planar sheets have equivalent ultrastructure, with the 54 Å repeat representing a key building block in polymer assembly.

## Vipp1Δα6$_{1–219}$ truncation forms tightly packed planar spirals and highly ordered sheets

Truncation of the Vipp1 CTD (termed Vipp1Δα6$_{1–219}$) modulates Vipp1 polymerization dynamics. Specifically, Vipp1Δα6$_{1–219}$ purified in low-salt conditions (10 mM NaCl), has a greater propensity to form ordered helical filaments and quasi-helical ring stacks than does Vipp1 (Extended Data Fig. 3a)[1]. We therefore investigated how removal of the CTD would affect Vipp1 behavior on SLBs and lipid monolayers. When mixed with SLBs and visualized by F-AFM, Vipp1Δα6$_{1–219}$ bound the SLB edges like

Vipp1. However, edges were coated with thin curled sheets or compact spirals where turns were usually interconnected and no central ring formed (Extended Data Fig. 3b). Both spiral filaments and planar sheets had height offsets of ~6 nm from the membrane, as Vipp1 did (Extended Data Fig. 3c). Remarkably, Vipp1Δα6$_{1–219}$ had the capacity to form crystalline planar sheets up to a micrometer in width, with surface ridges 54 Å apart (Extended Data Fig. 3d). On the basis of these dimensions, we concluded that the ultrastructure of Vipp1Δα6$_{1–219}$ spirals and planar sheets is closely related to that of Vipp1. Overall, truncation of the CTD reduced filament dynamics, resulting in spirals merging and a tendency to form sheets.

Importantly, detailed analysis of Vipp1Δα6$_{1–219}$ sheets showed how they curl and yielded important insights into the rules that govern Vipp1 polymer curvature. When filaments curl, the rigid 54 Å-spaced substructure experiences tension or compression on the longer outside or shorter inside of the bend, respectively. Owing to the width of the sheets, curling occurs by the addition of filament sections into the outside of bends (wedging) while sections are removed on the inside (Extended Data Figs. 3e,f). This highlights that flexibility between the 54 Å-spaced ridges is limited, allowing only a defined squeezing or stretching of the spacing. This rule extends to the thinner spiral filaments, which are also governed by the 54 Å repeating substructure. Here, filament curvature is seldom induced by wedging (Fig. 4b). Instead, filaments achieve high curvature either by inducing lattice breaks or by tilting and transitioning to 3D structures to alleviate elastic stress[41,48].

Subsequent analysis of Vipp1Δα6$_{1–219}$ with lipid monolayers revealed rod-like filament structures bound to the surface that were transitioning to planar-like arrays (Extended Data Fig. 3g). In the background, a mosaic of planar filaments corralled around apparent raised regions and areas with ruptured monolayers (Extended Data Fig. 3h). Consistent with the F-AFM data, filaments merged into larger sheets, and distinct spirals were not readily observed (Extended Data Fig. 3i). Notably, both 54 Å spacing and orthogonal 32 Å spacing were detected, with the latter consistent with the axial rise between subunits in neighboring rungs of Vipp1 rings[1]. Overall, our data support a model wherein Vipp1Δα6$_{1–219}$ forms planar polymers with the same substructure as Vipp1. The CTD modulates Vipp1 polymer stability and dynamism and in its absence, Vipp1 has a higher avidity to form polymers with increased regularity and stability.

## Vipp1 helical filaments have a lattice closely related to Vipp1 rings$_{HS}$

Given that two-dimensional (2D) planar sheets share a close geometric relationship with helical lattices[54], and as a means of understanding the interconnectedness of Vipp1 polymers including rings, spirals and planar sheets, we determined the structure of four Vipp1 helical filaments by cryo-EM (Table 1). Specifically, Vipp1 and a Vipp1 interface 3 mutant, Vipp1-F197K/L200K[1], were assembled using equivalent helical parameters and lattices (Extended Data Fig. 4) and were termed Vipp1$_{L1}$ and Vipp1$_{F197K/L200K\_L1}$, respectively. Alternatively, Vipp1Δα6$_{1–219}$ assembled with related but different lattices, termed Vipp1$_{Δα6\_L2}$ and Vipp1$_{Δα6\_L3}$. Vipp1Δα6$_{1–219}$ formed longer and more stable helical filaments than native Vipp1 (ref. [1]), with helical-like ribbons and rings also observed

**Fig. 3 | Vipp1 spirals form protruding central rings that abscise. a**, F-AFM phase timecourse showing ring biogenesis from an exponential-shaped spiral (see Extended Data Figure 1a). Scan rate, 150 Hz; 256 × 213 pixels. **b**, F-AFM timecourse showing how spiral and ring maturation correlates with increased filament offset from the membrane. Blue and red dashed lines indicate plotted height profile. Blue and red arrows indicate equivalent positions between AFM images and in-plane distance plot. Scan rate, 20 Hz; 256 × 183 pixels. **c**, Quantification of height difference between Vipp1 rings$_{LS}$ and surrounding spiral filaments. Data were derived from n = 16 independent measurements. Error bars show one s.d. of the mean. **d**, Schematic showing spiral and ring biogenesis pathways. **e**, Single frame (left) and F-AFM height images averaged together

(right), showcasing the different types of Vipp1 ring observed. For Rings$_{LS}$, the height scale reflects the difference between the central ring and neighboring spiral. **f**, Quantification of Vipp1 ring diameters, similar to those in **e**. Data derived from n = 20, 39, and 39 independent measurements for Rings$_{HS}$, Rings$_{LS}$, and Rings (no spiral), respectively. Error bars show one s.d. of the mean. **g**, F-AFM phase image in which Vipp1 rings$_{HS}$ scan and stably bind highly curved or ruptured membrane. The zoomed-in area is enclosed in the dashed box, with a 40 nm scale bar. Scan rate, 15 Hz; 256 × 151 pixels. **h**, Quantification of Vipp1 rings$_{HS}$ (red line) and rings$_{LS}$ (blue line) height profiles show similar shape and lateral dimensions. Error bands show one s.d. of the mean.

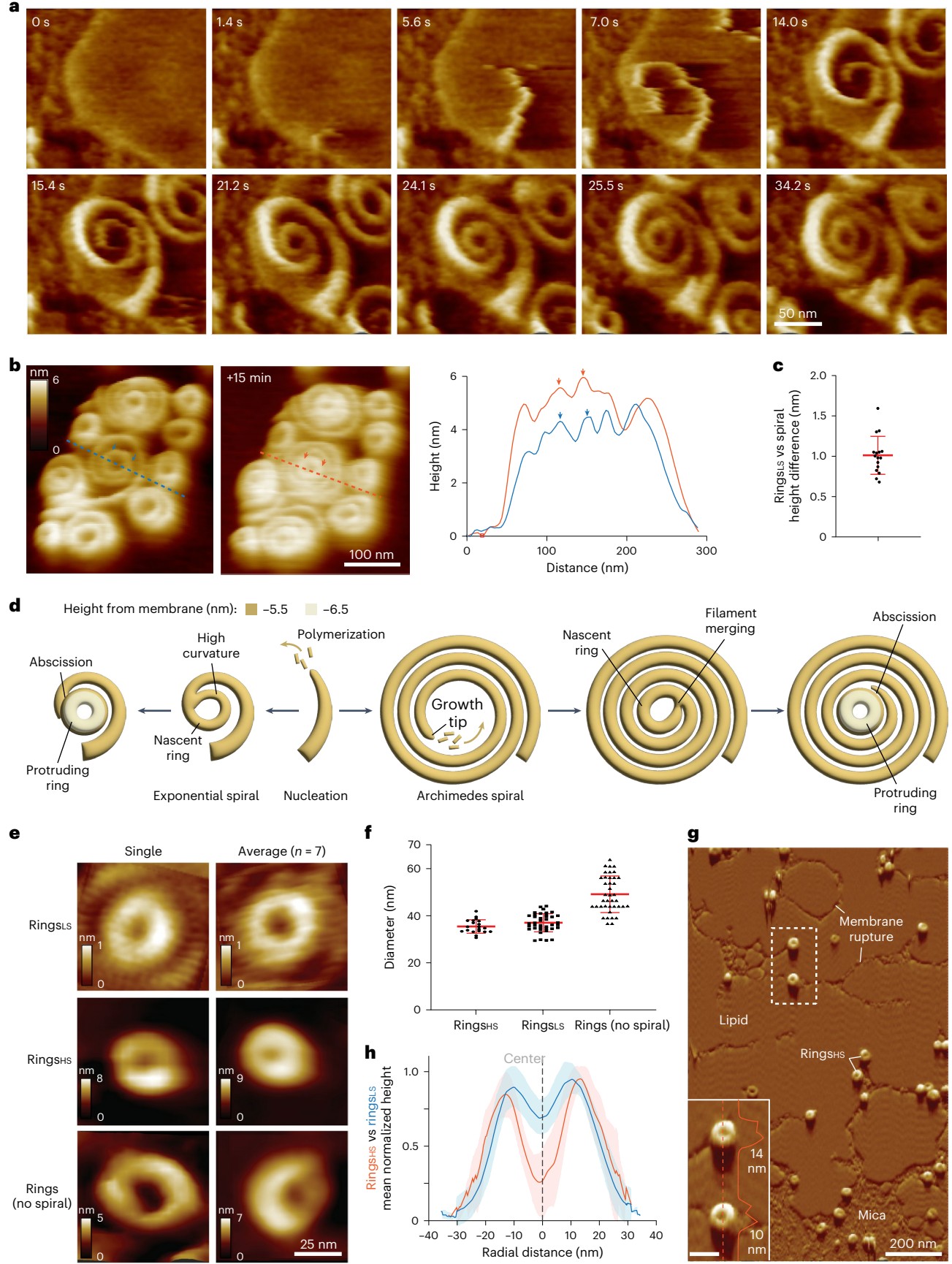

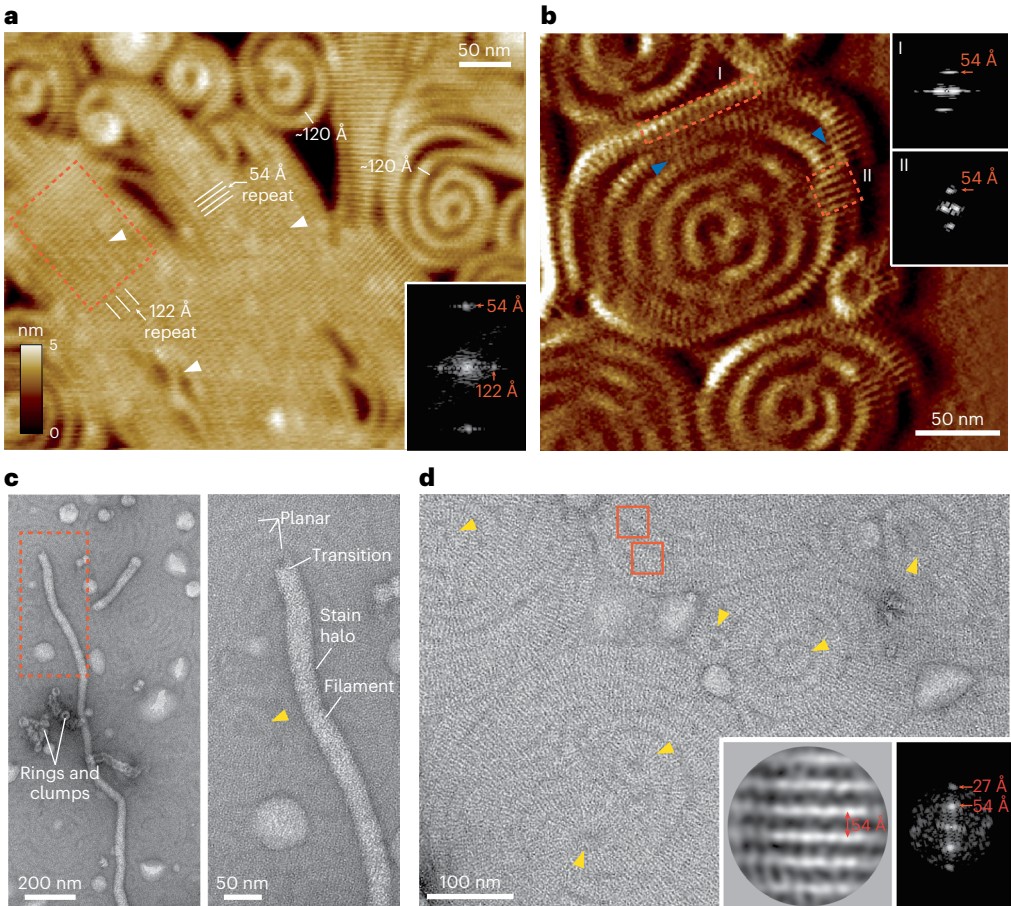

**Fig. 4 | Vipp1 planar sheets and spiral filaments have closely related lattices.**
**a**, F-AFM height image showing highly ordered planar sheets in Vipp1. Inset, Fourier Transform of the region enclosed by the dashed box. Parallel ridges are spaced 54 Å apart. White arrows indicate merged filament seam lines with 122 Å repeat. Scan rate, 10 Hz; 256 x 151 pixels. **b**, F-AFM phase image showing Vipp1 spirals. Insets, Fourier transform of the regions enclosed by the dashed boxes. Parallel ridges are spaced 54 Å apart. Blue arrows indicate the merging of spiral turns into a planar sheet, showcasing their close polymeric relationship. Scan rate, 35 Hz; 256 × 213 pixels. **c**, NS EM image showing Vipp1 polymers (rings and rod-like filaments) decorating the surface of a monolayer, which is itself covered by Vipp1 planar filaments. The experiment was repeated independently three times. **d**, NS EM image showcasing the mosaic of 2D planar spirals and sheets Vipp1 forms on a lipid monolayer. Yellow arrows indicate rings at the center of spirals. Red boxes indicate example regions for particle extraction and alignment. Inset, particle class average (left) and corresponding Fourier transform (right). Filament stripes are 54 Å apart. The experiment was repeated independently three times.

(Fig. 5a and Supplementary Fig. 1a). In all samples, helical filaments were heterogeneous, with multiple symmetries that required in silico classification to achieve near-uniform symmetry bins. Helical parameters could not be determined from low-quality $C_1$ symmetry reconstructions. Vipp1$_{\Delta\alpha6\_L3}$ was the only lattice in which class averages of aligned particles yielded a Fourier Transform with non-overlapping layer lines amenable to indexing (Extended Data Fig. 4). This produced a grid of possible symmetries that were systematically tested. Only helical parameters with a rise of 2.16 Å and a rotation of 85.50° yielded a 3.7 Å-resolution map (Fig. 5b and Extended Data Fig. 4a), with excellent side chain detail enabling model building from amino acids 1–217 (Fig. 5c). Map quality was lowest around the C terminus of helix 5 (α5C), indicative of instability within interface 1—a trait observed for all Vipp1 helical filaments, to differing degrees. Vipp1$_{\Delta\alpha6\_L3}$ was 24.4 nm in diameter, with a hollow inner lumen that was 12.7 nm across. Overall, the Vipp1 subunits self-assembled using similar interfaces to those used in Vipp1 ring$_{HS}$ polymerization[1], indicating a close relationship between helical and ring polymers (Fig. 5d–f). Specifically, Vipp1 formed ESCRT-III-like protofilaments (Fig. 5g–i), in which hairpin motifs packed side by side and the α5C of subunit $j$ bound across the hairpin tip of subunit $j$+3 to form the classical interface 1 domain swop (Fig. 5d,e). These protofilaments formed a 17-start right-handed helix that ran diagonally around the helical axis (Extended Data Fig. 4b). Concurrently, each hairpin tip bound the N terminus of helix 5 (α5N) in a neighboring protofilament, thereby forming interface 3 (Fig. 5d)[1]. Neighboring protofilaments aligned laterally to form a 4-start left-handed helix, in which subunits connected through the stacking of helices α0 and the formation of interface 2 (Fig. 5f)[1]. Helix α0 lipid-binding domains were therefore lined and twisted around the inner lumen of the filament. The 4-start helix has a pitch of 44 Å and forms diagonal parallel ridges that encircle the helical axis and dominate the surface topology of the Vipp1 helical filament. The structure of Vipp1$_{\Delta\alpha6\_L2}$ was very similar to that of Vipp1$_{\Delta\alpha6\_L3}$, although different helical parameters yielded a modified helical lattice with a slight lattice rotation relative to the helical axis (Extended Data Fig. 5a–c). Overall, the Vipp1$_{\Delta\alpha6\_L3}$ and Vipp1$_{\Delta\alpha6\_L2}$ structures revealed the close relationship between Vipp1 helical and ring$_{HS}$ polymers, highlighting how relatively minor adjustments to assembly dynamics facilitate transition between these polymer types.

## Comparison of Vipp1$_{L1}$ with Vipp1$_{\Delta\alpha6\_L2}$ and Vipp1$_{\Delta\alpha6\_L3}$ revealed a mechanism for filament constriction
Compared with the Vipp1Δα6$_{1-219}$ sample, Vipp1 was characterized by unfurled ring stacks or helical-like ribbons with fewer and shorter ordered helical filaments, suggesting a reduced stability for helical

**Table 1 | Cryo-EM data collection, refinement and validation statistics**

| | Vipp1$_{L1}$ (EMD-18318), (PDB 8QBR) | Vipp1$_{F197K/L200K\_L1}$ (EMD-18319), (PDB 8QBS) | Vipp1$_{\Delta\alpha6\_L2}$ (EMD-18321), (PDB 8QBV) | Vipp1$_{\Delta\alpha6\_L3}$ (EMD-18322), (PDB 8QBW) |
|---|---|---|---|---|
| **Data collection and processing** | | | | |
| Magnification | ×81,000 | ×81,000 | ×81,000 | ×81,000 |
| Voltage (kV) | 300 | 300 | 300 | 300 |
| Electron exposure (e⁻/Å²) | 50 | 50 | 41 | 41 |
| Defocus range (µm) | 0.75–2.5 | 0.75–2.5 | 0.75–2.5 | 0.75–2.5 |
| Pixel size (Å) | 1.072 | 1.1 | 1.1 | 1.1 |
| Symmetry imposed | $C_1$ and helical | $C_1$ and helical | $C_1$ and helical | $C_1$ and helical |
| Initial particle stack | 138,562 | 1,172,337 | 648,059 | 648,059 |
| Final particle stack | 43,480 | 36,652 | 38,361 | 38,585 |
| Helical rise (Å) | 2.372 | 2.440 | 2.155 | 2.159 |
| Helical twist (°) | −75.860 | −75.835 | 68.507 | 85.495 |
| Map resolution (Å) | 3.7 | 3.7 | 3.8 | 3.7 |
| FSC threshold | 0.143 | 0.143 | 0.143 | 0.143 |
| Map resolution range (Å) | 3–6 | 3–5 | 3–5 | 3–5 |
| **Refinement** | | | | |
| Initial model used | 8QBW (this work) | 8QBW | 8QBW | 6ZW4 |
| CC mask† | 0.60 | 0.74 | 0.52 | 0.72 |
| Model resolution (Å)† | | | | |
| FSC threshold (0.143) | 3.7 | 3.3 | 3.9 | 3.5 |
| FSC threshold (0.5) | 7.0 | 3.9 | 7.4 | 3.9 |
| Map sharpening $B$ factor (Å²) | 91.1 | −110.7 | 189.0 | 183.9 |
| Model composition | | | | |
| Non-hydrogen atoms | 1,671 | 1,570 | 1,693 | 1,693 |
| Protein residues | 214 | 201 | 217 | 217 |
| Ligands | N/A | N/A | N/A | N/A |
| $B$ factors (Å²) | | | | |
| Protein (min/max/mean) | 29.4/256.6/112.9 | 29.6/190.8/73.3 | 18.5/82.4/45.9 | 5.7/176.5/56.4 |
| Ligand | N/A | N/A | N/A | N/A |
| R.m.s. deviations | | | | |
| Bond lengths (Å) | 0.005 | 0.007 | 0.008 | 0.004 |
| Bond angles (°) | 1.118 | 0.960 | 1.206 | 0.861 |
| Validation | | | | |
| MolProbity score | 1.71 | 1.17 | 1.77 | 1.11 |
| Clashscore† | 17.05 | 8.89 | 21.46 | 8.42 |
| Poor rotamers (%) | 0.00 | 0.00 | 0.00 | 0.00 |
| Ramachandran plot | | | | |
| Favored (%) | 97.17 | 98.48 | 97.21 | 99.07 |
| Allowed (%) | 2.83 | 1.52 | 2.79 | 0.93 |
| Disallowed (%) | 0.00 | 0.00 | 0.00 | 0.00 |

†Computed for a biological/filament assembly: 95-base polymer, 5-start turn for Vipp1$_{L1}$ and Vipp1$_{F197K/L200K\_L1}$ 105-base polymer, 5-start turn for Vipp1$_{\Delta\alpha6\_L2}$ 84-base polymer, 4-start turn for Vipp1$_{\Delta\alpha6\_L3}$ N/A, not applicable.

polymerization (Extended Data Fig. 6a and Supplementary Fig. 1b). Although the Fourier transforms of Vipp1$_{L1}$ and Vipp1$_{\Delta\alpha6\_L2}$ class averages were similar with closely related lattices, only helical parameters with a rise of 2.37 Å and a rotation of −75.86° yielded a Vipp1$_{L1}$ map at an overall resolution of 3.7 Å (Extended Data Figs. 4 and 6b). Map local resolution range was wider than other Vipp1 helical forms with the inner parts of the filament well resolved compared with the periphery, where the hairpin tips and helix α5 were more poorly ordered (Extended

Data Fig. 6c). The Vipp1$_{L1}$ subunit was built between amino acids 1–214 (Extended Data Fig. 6d,e) and assembled using similar interfaces as for Vipp1$_{\Delta\alpha6\_L2}$ and Vipp1$_{\Delta\alpha6\_L3}$. However, the Vipp1$_{L1}$ subunit differed from the Vipp1$_{\Delta\alpha6\_L2}$ and Vipp1$_{\Delta\alpha6\_L3}$ subunits, with its hairpin tip kinked at Ile68 and helix α5 compacted (Extended Data Fig. 6d,e) such that interfaces 1 and 3 were perturbed. Although map relating to the CTD could not be assigned, our structures were consistent with this motif destabilizing interface 1, possibly through kinking of the hairpin while directly or

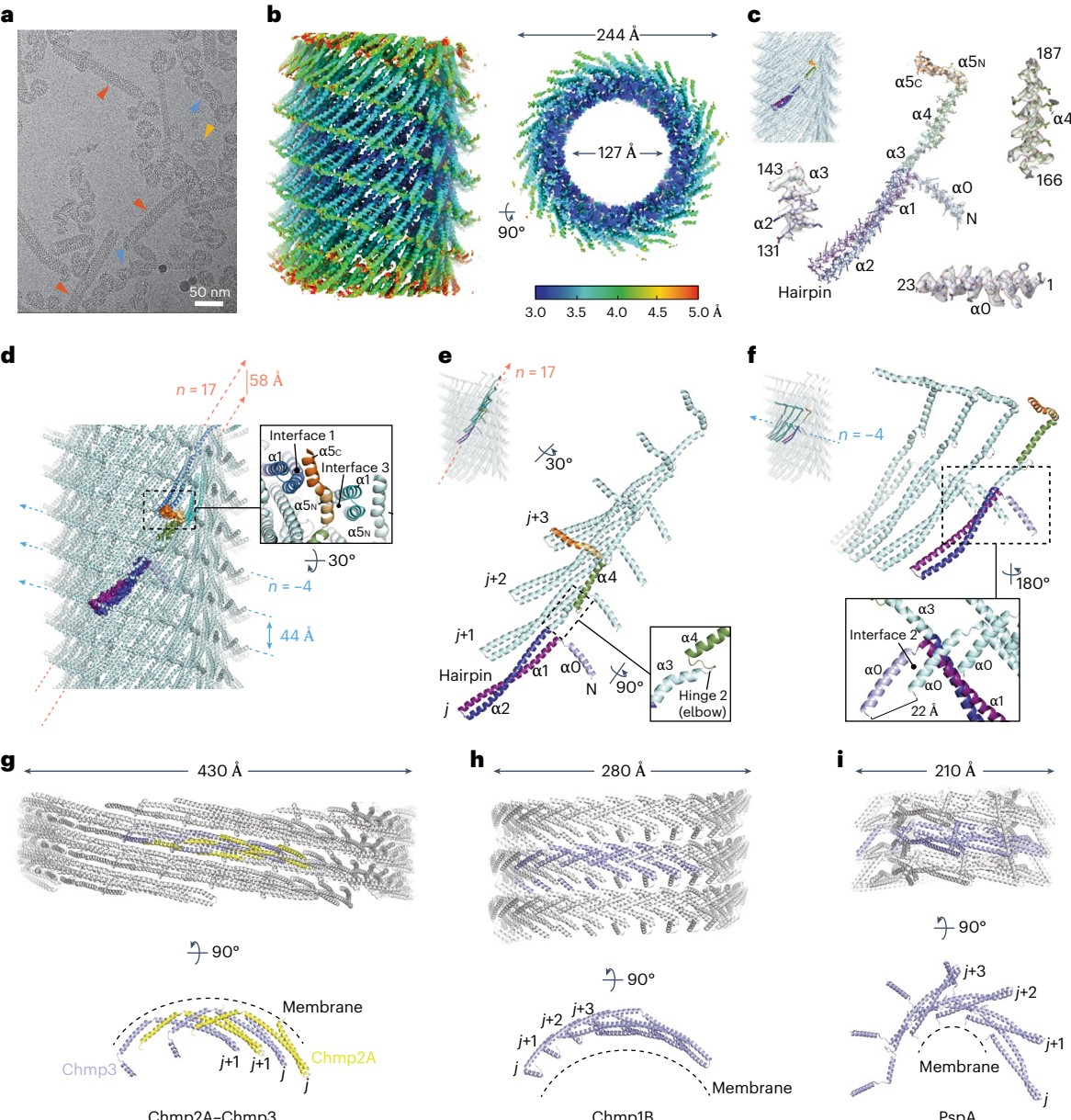

**Fig. 5 | Vipp1Δα6$_{1-219}$ helical filaments have a lattice closely related to Vipp1 rings$_{HS}$. a**, Cryo-EM image showing Vipp1Δα6$_{1-219}$ forming helical filaments, helical-like ribbons, and rings (red, blue, and yellow arrows, respectively). A zoomed version is shown in Supplementary Data Figure 1a. The experiment was repeated independently three times. **b**, Sharpened Vipp1$_{Δα6\_L3}$ map contoured at 2.3σ, showing local-resolution estimates. **c**, Vipp1$_{Δα6\_L3}$ map fitted with Vipp1$_{Δα6\_L3}$ helical filament structure (top left). The colored monomer is isolated and zoomed to show map quality, build, and fit. The map is contoured at 3σ, except for helix α5 at 1σ. **d**, Structure of the Vipp1$_{Δα6\_L3}$ helical filament; the zoomed panel highlights conservation of interfaces 1 and 3. Bessel orders $n = 17$ and $n = -4$ are indicated with pitch. **e**, The 17-start right-handed helix in Vipp1$_{Δα6\_L3}$ forms ESCRT-III-like protofilaments. **f**, The 4-start left-handed helix in Vipp1$_{Δα6\_L3}$ is formed by subunit stacking mediated by interface 2. **g**–**i**, Helical structures of other ESCRT-III family members bound to membrane.

indirectly limiting the formation of interface 3. The effect of the CTD on interfaces 1 and 3 thereby present a mechanism for Vipp1$_{L1}$ helical filament instability and tuning of Vipp1 polymer dynamics[1,31,32]. Compared with Vipp1$_{Δα6\_L2}$ and Vipp1$_{Δα6\_L3}$, the Vipp1$_{L1}$ filament was constricted with a 21.0 nm and 10.5 nm external and inner diameter, respectively. Subunit removal was the mechanism underlying this ~2 nm inner diameter constriction. Specifically, the left-handed 21-start helix in Vipp1$_{Δα6\_L2}$ and Vipp1$_{Δα6\_L3}$ that runs nearly parallel to the helix axis formed a 19-start helix in Vipp1$_{L1}$ (Extended Data Fig. 4b). Filament constriction was mediated by angular adjustments of helices α3 and α4, resulting in a downward flexing of ~7.5 Å (Extended Data Fig. 6e). These intra-subunit conformational changes were accompanied by subtle adjustments in subunit lattice packing, resulting in a Vipp1$_{L1}$ protofilament with higher curvature than Vipp1$_{Δα6\_L2}$ or Vipp1$_{Δα6\_L3}$ (Extended Data Fig. 6f and Supplementary Video 4). Overall, the comparison of Vipp1$_{L1}$, Vipp1$_{Δα6\_L2}$, and Vipp1$_{Δα6\_L3}$ filaments provided a mechanism for constriction wherein CTD-mediated adjustments to lattice assembly induced a reduction of ESCRT-III-like protofilaments and consequently a reduction in filament circumference. Our data are consistent with a model in which the CTD plays a key role in tuning Vipp1 assembly dynamics and has the capacity to drive helical filament constriction.

## Vipp1 polymers tubulate membrane along the same plane from different orientations

The Vipp1 F197K and L200K mutations impeded interface 3, resulting in long helical filaments[1] that often had a membrane vesicle cap at their

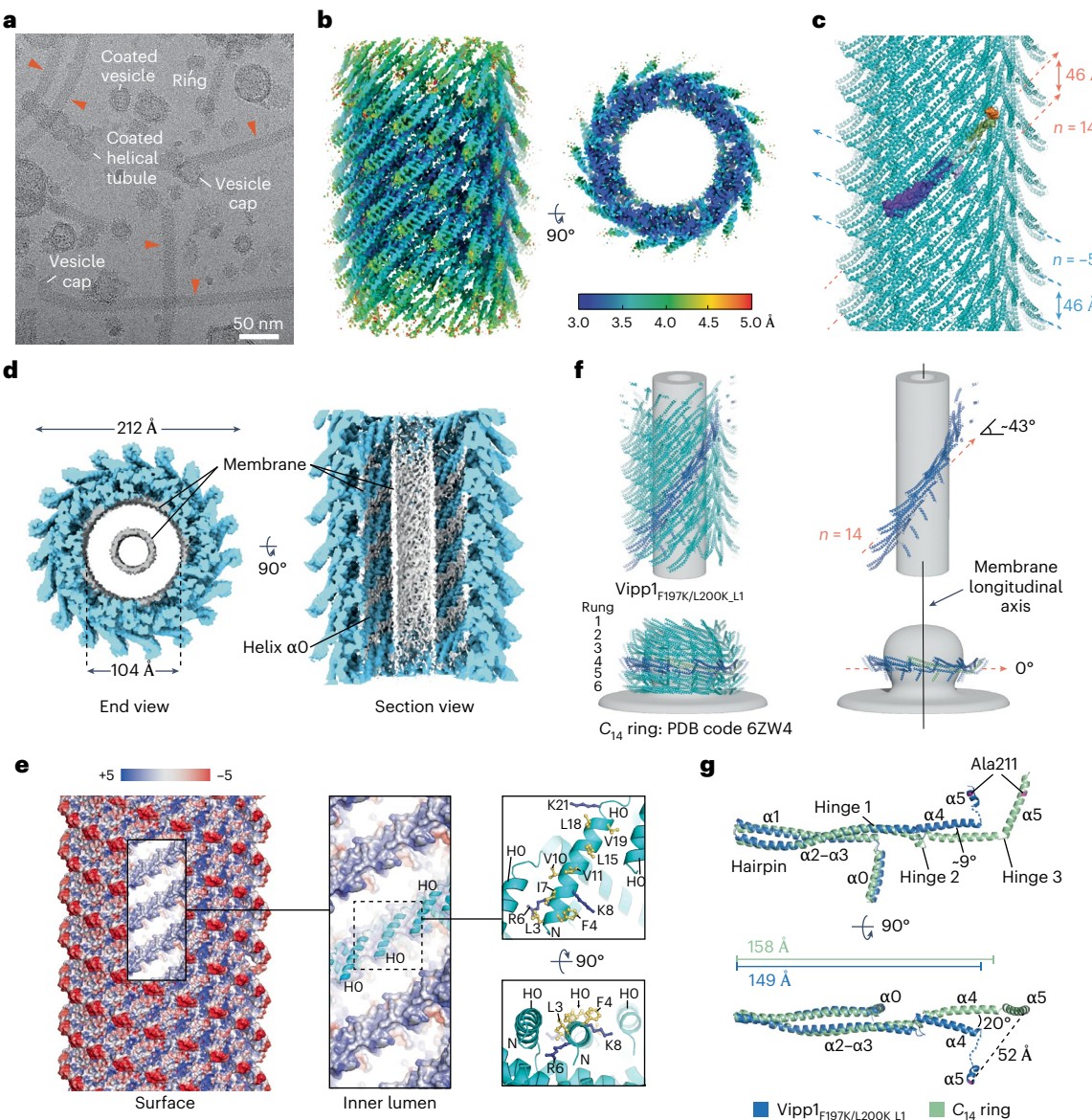

**Fig. 6 | Vipp1$_{F197K/L200K\_L1}$ is constricted and tubulates membrane. a**, Cryo-EM image showing Vipp1$_{F197K/L200K\_L1}$ forming helical filaments and coated membrane tubules. A zoomed version is shown in Supplementary Data Figure 1b. The experiment was repeated independently three times. **b**, Sharpened Vipp1$_{F197K/L200K\_L1}$ map contoured at 4σ showing local resolution estimates. **c**, Structure of the Vipp1$_{F197K/L200K\_L1}$ helical filament with one monomer colored. Bessel orders $n = 14$ and $n = -5$ are indicated with pitch. **d**, Unsharpened Vipp1$_{F197K/L200K\_L1}$ map

contoured at 3σ, showcasing tubulated membrane within the inner lumen. **e**, Vipp1$_{F197K/L200K\_L1}$ filament surface rendered to show electrostatic charge. The blue to red spectrum represents positive to negative charges with units $k_BT/e_c$. Zoomed panels show the mechanism of membrane binding. **f**, Comparison of ESCRT-III-like protofilament orientation relative to the same membrane plane between Vipp1$_{F197K/L200K\_L1}$ and Vipp1 ring$_{HS}$. **g**, Hairpin superposition of a Vipp1$_{F197K/L200K\_L1}$ subunit with a Vipp1 ring$_{HS}$ subunit ($C_{14}$ symmetry, rung 4, PDB code 6ZW4).

tips (Fig. 6a and Supplementary Fig. 1c). Spheroid and helical tubule-like membrane vesicles coated in Vipp1$_{F197K/L200K}$ were also observed. The membrane was presumably bound during purification from *Escherichia coli*. Vipp1$_{F197K/L200K\_L1}$ was resolved to 3.7 Å overall (Fig. 6b and Extended Data Fig. 4a) using helical parameters of a rise of 2.44 Å and a rotation of −75.83°, which were almost identical to those of Vipp1$_{L1}$. Excellent map quality and side chain detail facilitated model building (Extended Data Fig. 5d). Notably, this included flexible hinge 2, which was well ordered compared with other Vipp1 maps. Overall, Vipp1$_{F197K/L200K\_L1}$ and Vipp1$_{L1}$ subunits were in similar conformations (Extended Data Fig. 5e) and assembled using equivalent helical lattices (Fig. 6c and Extended Data Fig. 4). However, owing to the Vipp1$_{F197K/L200K}$ mutation, interface 3 formation was inhibited, with no supporting map and build for helix α5N. No supporting map was located for the CTD. The key difference between the Vipp1$_{F197K/L200K\_L1}$ and Vipp1$_{L1}$ maps was the presence of lipid bilayer

in the central lumen of Vipp1$_{F197K/L200K\_L1}$. Here, the inner leaflet formed a tube with a diameter of 4 nm, close to the limit at which hemifusion is expected to occur. Concurrently, the outer leaflet filled the space between the helix α0 stacks (Fig. 6d). Analysis of the Vipp1$_{F197K/L200K\_L1}$ electrostatic surface potential showed that amphipathic helix α0 was positively charged, with basic residues positioned to attract negatively charged lipid headgroups. Hydrophobic residues oriented along the inner lumen face then contacted the fatty acid chains (Fig. 6e). Notably, the structure of Vipp1$_{F197K/L200K\_L1}$ facilitated a comparison of Vipp1 helical and ring$_{HS}$ polymers when bound to membrane. In the Vipp1 ring$_{HS}$ with $C_{14}$ symmetry, the ESCRT-III-like protofilaments formed rungs that were orthogonal to the membrane bud–tube axis[1]. In Vipp1$_{F197K/L200K\_L1}$, the ESCRT-III-like protofilaments were rotated ~43° to the membrane tube axis while maintaining similar lateral interactions and ultrastructure by undergoing filament twisting (Fig. 6f). Therefore, by forming a 3D

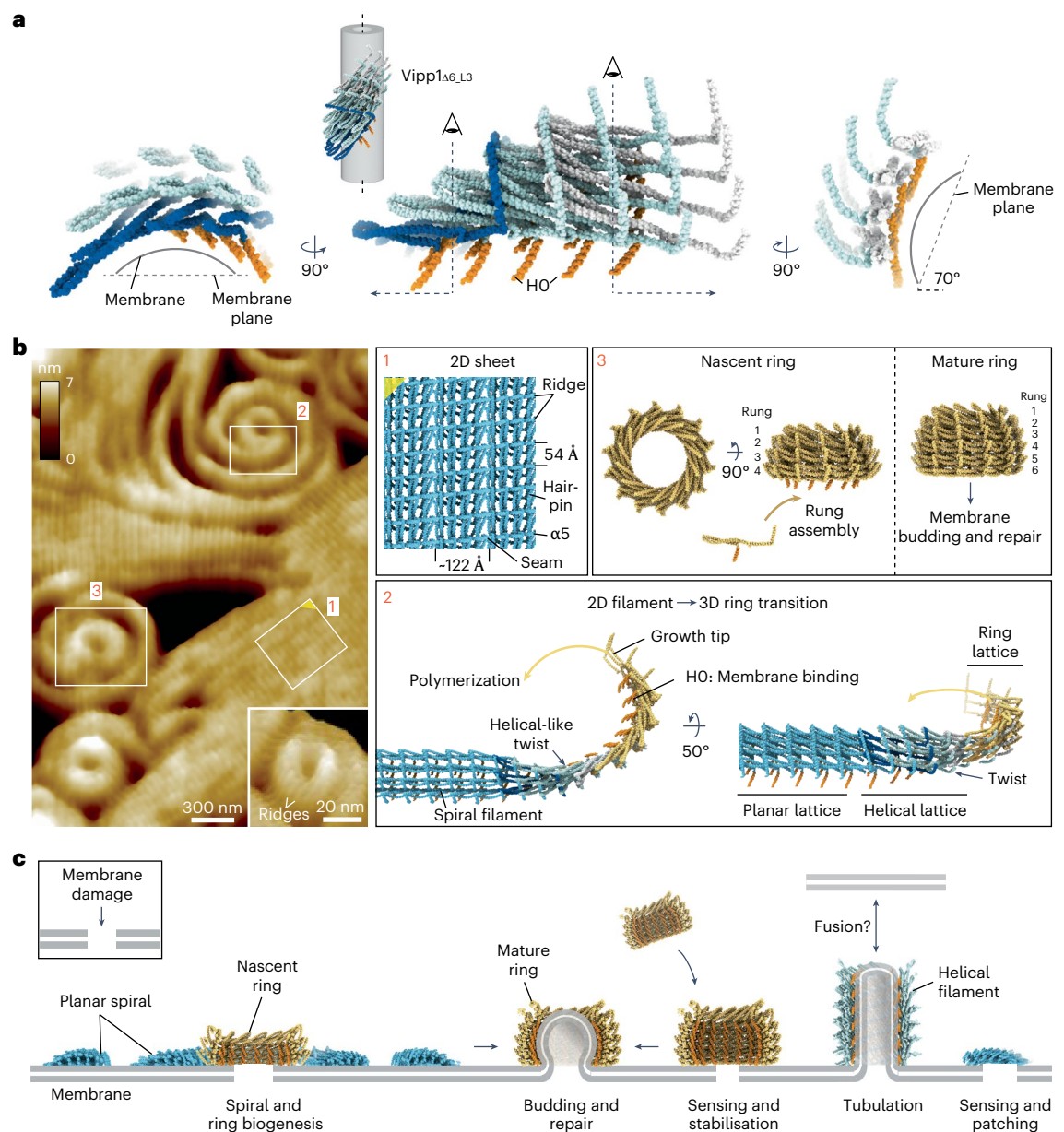

**Fig. 7 | Mechanism for Vipp1 spiral formation, ring biogenesis, and membrane repair. a**, Section of four ESCRT-III-like protofilaments extracted from the Vipp1$_{\Delta\alpha6\_L3}$ helical structure, showing how filament twist enables binding of membranes on opposing planes. **b**, Model of Vipp1 planar sheet, spiral, and 3D ring biogenesis. F-AFM image (left); the white boxes labeled 1–3 are shown in the panels at the right labeled 1–3. Scan rate, 10 Hz; 256 × 151 pixels. The twisting helical filament in **a** bridges the transition from 2D planar to 3D ring structures. To facilitate modeling of the planar filament, helix α4 was removed. **c**, Mechanism of Vipp1-mediated membrane sensing, stabilization, and repair. In *Chlamydomonas*, Vipp1 helical filaments may bridge thylakoids and the chloroplast envelope[35].

pliable polymer, the Vipp1 lattice can induce membrane tubulation along the same axis but by using radically different lattice orientations. To facilitate the transition from Vipp1 ring$_{HS}$ to helical polymer, substantial conformational changes were observed within individual subunits to accommodate the relative shift in membrane plane (Fig. 6g and Supplementary Video 5). Mediated by hinges 1 and 2, helices α3 and α4 flexed upwards and outwards ~9° and ~20°, respectively. Hinge 3 facilitated additional shifts in helix α5C orientation, culminating in a ~52 Å swing as measured by Ala211 movement. In addition, the length of the Vipp1 subunit was reduced from 158 Å in Vipp1 ring$_{HS}$ to 149 Å in Vipp1$_{F197K/L200K\_L1}$ between the hairpin tip and helix α4 terminus. This important compaction was mediated by hinge 1 with the N terminus of helix α4 overlapping the C terminus of helix α3. Overall, Vipp1$_{F197K/L200K\_L1}$ showed how *Nostoc punctiforme* Vipp1 helical filaments

have the capacity to tubulate membrane, as reported in other Vipp1 systems[55]. Moreover, it shows that the transition between helical and ring conformations relative to the membrane plane is induced by lattice rotation and filament twist rather than through a novel polymer form. This is important because the transition of ESCRT-III planar spirals and rings to 3D spirals likely requires filament rotation relative to the membrane plane, as previously observed[27,34,51].

## Discussion

Here we show how the bacterial ESCRT-III-like protein Vipp1 assembles dynamic planar spirals and sheets on membrane. Moreover, the spirals assemble central rings that protrude and abscise from the parent filament. These rings have similar dimensions to Vipp1$_{HS}$ rings known to bud membrane. In addition, we show the structure of four Vipp1

**Table 2 | Cryo-ET data collection parameters**

| | Vipp1Δα6$_{1-219}$ |
|---|---|
| **Data collection and processing** | |
| Magnification | ×35,445 |
| Voltage (kV) | 300 |
| Energy filter slit width (eV) | 20 |
| Electron exposure per tilt (e⁻/Å²) | 2.6 |
| Total electron dose (e⁻/Å²) | 102 |
| Defocus range (μm) | 3–6 |
| Tilt range (degrees) | ±60 |
| Tilt step (degrees) | 3 |
| Acquisition scheme | Dose-symmetric |
| Pixel size | 2.257 Å |
| Symmetry imposed | N/A |
| Initial subtomograms (no.) | N/A |
| Final subtomograms (no.) | N/A |
| Map resolution (Å) | N/A |
| FSC threshold | N/A |
| Map resolution range (Å) | N/A |

helical filaments with three related but distinct lattice assemblies. These structures show the close connection between 2D planar, helical, and Vipp1$_{HS}$ ring polymers. Given that these different polymer forms are highly ordered, their comparison provides a basis for modeling how Vipp1 builds planar filaments that transition to 3D membrane budding forms through filament twist (Fig. 7a).

Both F-AFM and NS EM data show Vipp1 planar sheets and filaments that are characterized by parallel ridges with a 54 Å repeat (Fig. 4a,b,d). Additionally, the Vipp1Δα6$_{1-219}$ planar filaments include a 32 Å repeat orthogonal to the 54 Å spacing (Extended Data Fig. 3i). By calculating the cylindrical projection[56] of the Vipp1$_{Δα6\_L3}$ helical filament (Extended Data Fig. 7a), the 3D map may be represented as a geometrically equivalent 2D lattice. The 4-start left-handed helix is the dominant feature forming parallel stripes with 44 Å pitch. These stripes relate to the ridges on the Vipp1$_{Δα6\_L3}$ filament surface when visualized by cryogenic electron tomography (Extended Data Fig. 7b and Table 2). The inter-ridge distance is therefore formed by the spacing between neighboring hairpins in each ESCRT-III-like filament (Extended Data Fig. 7c). Intriguingly, inter-hairpin distance has a degree of flexibility, with hairpins sliding up to 14 Å relative to each other in Vipp1 rings$_{HS}$, depending on rung position (Extended Data Fig. 7d). In Vipp1 rings$_{HS}$ with $C_{17}$ symmetry, hairpin spacing spans 53–59 Å in the central rungs with each neighboring ESCRT-III-like filament 32–33 Å apart (Extended Data Fig. 7c). Overall, the lattice dimensions of helical and ring$_{HS}$ polymers are consistent with those obtained from the Fourier transform of Vipp1 and Vipp1Δα6$_{1-219}$ planar sheets and filaments on a lipid monolayer (Fig. 4d and Extended Data Fig. 3i) and SLBs (Fig. 4a,b). Our data therefore support a model in which Vipp1 planar sheets and spiral filaments are geometrically similar to unfurled and flattened Vipp1 rings$_{HS}$ or helical filaments, with the key assembly Interfaces 1–3 maintained. Sheets and spiral filaments comprise parallel ESCRT-III-like protofilaments (Fig. 5e), with the ridges running near orthogonal to the protofilament axis (Fig. 7b). The 122 Å spacing observed as seams in the sheets is close to the ~13 nm width of spiral filaments, measured by F-AFM (Fig. 2g) and is consistent with spiral filaments comprising four, or sometimes five, parallel protofilaments. Snf7 spirals generally form from just one protofilament, which likely explains their increased compressibility and deformation into polygons[41].

As Vipp1 spiral filaments grow centrally, curvature increases with each turn, building elastic stress[41,48]. For spiral filaments to curve, our data support a model wherein hairpin sliding and subunit flexing through hinges 1–3 enable each planar filament to curl laterally on the membrane plane, with the inter-ridge distance shortening on the inside of the filament and extending on the outside. However, on the basis of measurements from different rungs in Vipp1 rings$_{HS}$[1], the inter-ridge distance cannot compress below ~41 Å or stretch greater than ~61 Å (Extended Data Fig. 7d). These distances are insufficient to facilitate the high level of curvature observed in the filaments or rings$_{LS}$ at the center of spirals if in a strictly planar form. Therefore, once inter-ridge distance limits are reached, residual curvature-induced elastic stress must be minimized, either by breaking the filament, by wedging (Extended Data Fig. 3f), or through filament tilt and transitioning into a 3D form. Here, filament tilt occurs as observed by an increase in filament height offset from the membrane, particularly within central turns of the spiral (Fig. 3c and Extended Data Fig. 1a).

In the center of the Vipp1 spirals, where the filament curvature was highest, Vipp1 rings$_{LS}$ formed that protruded an additional ~1 nm above the surrounding spiral filaments owing to filament tilt. Ridges were sometimes observable on the outside face of Vipp1 rings$_{LS}$ indicating filament tilt (Fig. 7b). Notably, the filament twist observed in Vipp1 helical polymers (Fig. 7a) provided a mechanism for how ESCRT-III-like filaments tilt and bind membrane on different planes, thereby enabling Vipp1 to transition from planar to 3D ring architectures (Fig. 7b). In ESCRT-III, changes in the geometry of membrane-bound filaments, including filament tilt, underlie membrane deformation from a planar spiral to a 3D helix[27,49,50]. Ultimately, filament tilt in Vipp1 rings$_{LS}$ relative to the parent spiral induces torsion and promotes abscission. Although Vipp1 rings$_{HS}$ have a substantially lower height when measured by F-AFM (Extended Data Fig. 2c) compared with structural measurements[1], our data were consistent with Vipp1 rings$_{LS}$ comprising ~3–4 rungs, rather than 5–7 rungs as in mature Vipp1 ring$_{HS}$ (Fig. 7b). Final maturation steps might require non-rigid membrane support, changes in membrane composition, or additional factors such as nucleotide hydrolysis[35,52].

Collectively, our results provide a mechanism for how Vipp1 functions in membrane stabilization and repair (Fig. 7c) in cyanobacteria and chloroplasts. Both Vipp1 small oligomers (or monomers) and Vipp1 rings$_{HS}$ sense and bind highly curved and perturbed membrane. This finding is consistent with Vipp1 localizing to the highly curved edges of thylakoid membranes[18]. Depending on the conditions, the small oligomers can polymerize into spirals that encircle the damaged region, which can incorporate protein complexes[19,57,58], ultimately assembling a central ring structure. Mature Vipp1 rings$_{HS}$ have the capacity to bud membrane, thereby supporting a repair mechanism based on membrane squeezing and fusion[1]. Alternatively, by tuning the CTD, spiral filaments can readily anneal to form crystalline concentric or linear sheets (Extended Data Fig. 3d), which would stabilize damaged membrane, acting as a physical barrier and supporting scaffold. Such structures explain how Vipp1 and other bacterial homologs such as PspA might inhibit proton leakage across membranes[47,59]. Finally, Vipp1 helical filaments may form membrane bridges linking thylakoids and the chloroplast envelope[35].

In conclusion, our results utilize Vipp1 as a system to show how planar filaments transition to 3D rings. The homology between Vipp1 and other ESCRT-III proteins suggests that the basic principles observed here, including conserved ultrastructure between planar and 3D forms, filament tilt and twist, and lattice rotation on an equivalent membrane plane, will extend to other family members.

## Online content

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

## Methods

### Expression and purification of Vipp1 filament and ring assembly

All Vipp1 clones originated from a previous study[1]. In brief, the coding sequences for *N. punctiforme vipp1* (Uniprot code B2J6D9), *vipp1Δα6*$_{1-219}$, and *vipp1*$_{F197K/L200K}$ were cloned into pOPTM (a pET derivative) to yield an N-terminal MBP fusion with a TEV cleavage site in the linker. An N-terminal hexa-histidine tag was included with the MBP moiety. For the purification of all Vipp1 clones, plasmids were co-transformed into *E. coli* C43 (DE3) electro-competent cells (Lucigen) that were modified to incorporate *pspA* gene knockout[60]. Cells were grown on LB-agar with ampicillin (100 µg ml$^{-1}$); 2×YT medium was inoculated, and the cultures were grown at 37 °C until reaching an optical density of 600 nm (OD$_{600}$) of 0.6, followed by induction with 1 mM isopropyl β-D-1-thiogalactopyranoside (IPTG). Cells were grown for 16 h at 19 °C and shaken at 200 r.p.m. The following steps were carried out at 4 °C, unless otherwise specified. Cell pellets were resuspended in buffer containing 50 mM HEPES-NaOH pH 7.5, 500 mM NaCl, 0.1 mg ml$^{-1}$ DNase-I, and Roche cOmplete EDTA-free protease inhibitor cocktail and sonicated on ice. The lysate was clarified by centrifugation at 16,000*g* for 20 min. The supernatant was incubated with 20 ml amylose resin (NEB) that was pre-equilibrated with lysis buffer for 15 min before loading onto a gravity flow column. Resin was washed with three column volumes (CV) of wash buffer (50 mM HEPES-NaOH pH 8.4, 200 mM NaCl) and two CV of ATP-wash buffer (50 mM HEPES-NaOH pH 8.0, 80 mM KCl, 2.5 mM MgCl$_2$, and 5 mM ATP), followed by two CV of wash buffer. Sample was eluted with wash buffer supplemented with 15 mM maltose. Peak fractions were pooled and incubated with TEV initially at 34 °C for 2 h, followed by 48 h at room temperature. Digested samples were dialyzed (12- to 14-kDa cut-off) overnight in size-exclusion chromatography (SEC) buffer containing 25 mM HEPES-NaOH pH 8.4, 10 mM KCl, and 10 mM MgCl$_2$, except for Vipp1Δα6$_{1-219}$ for which 25 mM HEPES-NaOH pH 8.4 and 10 mM NaCl was used. Samples were passed over a 10 ml Ni-NTA bead gravity flow column pre-equilibrated with SEC buffer to remove His-tagged MBP and TEV in the column. Vipp1 was collected from the flow through and concentrated using Vivaspin-20 concentrators with a 10-kDa cut-off. During this process, sample was iteratively diluted with SEC buffer before being concentrated to 5 ml. Samples were injected onto a Sephacryl 16/60 S-500 pre-equilibrated with SEC buffer. Gel filtration yielded three peaks. The first peak after the void volume at ~0.3 CV contained the Vipp1 helical polymers used in this study, the second peak at ~0.4–0.7 CV was predominantly populated with rings, and the third peak at ~0.75–0.9 CV comprised Vipp1 monomer or small oligomers, MBP, and TEV. For exclusive purification of Vipp1 rings$_{HS}$, a protocol using a SEC buffer with a high salt concentration (50 mM) was used, as described previously[1].

### Lipid-covered silica beads preparation

For fluorescent light microscopy (FLM) studies, lipid lamellae were deposited on 40 µm silica beads, as previously described[61]. In brief, 1,2-dioleoyl-*sn*-glycero-3-phosphocholine (DOPC; Avanti Lipids no. 850375), 1,2-dioleoyl-*sn*-glycero-3-phospho-L-serine (DOPS; Avanti Lipids no. 840035), and 1,2-dioleoyl-*sn*-glycero-3-phosphoethanolamine labeled with Atto 647N (Atto-647DOPE; Merck no. 42247) were mixed at a ratio of 59.95:40:0.05 mol%, respectively, from lipid stocks in chloroform to a final lipid concentration of 0.5 mg ml$^{-1}$. The lipid mixture was then dried in a vacuum for at least 2 h to completely remove the chloroform, forming a dried lipid film, followed by hydration and resuspension in a buffer containing 1 mM HEPES-NaOH at pH 7.4. Subsequently, 1 µl of 40 µm silica beads (Microspheres-Nanospheres no. 147070-10) was mixed with 10 µl of the hydrated lipid mixture and then divided into five drops placed on a parafilm surface. Subsequently, these bead–lipid drops were dried in a vacuum for at least 30 min until the complete evaporation of the aqueous buffer.

### Supported lipid bilayers preparation

For the FLM studies, SLBs were prepared as previously described[62]. In brief, coverslips were cleaned by sonication in water, ethanol, and water for 10 min each, followed by 30 s of plasma cleaning (Harrick Plasma). After plasma cleaning, an Ibidi chamber (sticky-Slide VI 0.4) was mounted on the coverslip, and each of the wells was filled with 200 µl of buffer (25 mM HEPES-NaOH, pH 8.3, 1 mM EDTA, and 10 or 500 mM NaCl depending on the experiment). To form the SLBs, a portion of the lipid-covered silica beads was transferred to each well using a glass pipette, leading to the spilling of the lipid bilayers on the coverslip. The fluorescently labeled protein was introduced in the assays by replacing 100 µl buffer with new buffer containing the protein at desired concentration in the final volume of 200 µl.

### Large unilamellar vesicles preparation

For F-AFM studies, LUVs were prepared using *E. coli* total lipid extract (Avanti, no. 100500C). This lipid, and the mix used for FLM, were tested with Vipp1, resulting in the assembly of similar polymers with sheets, spirals, and rings. Therefore, for the F-AFM results presented in this study, *E. coli* total lipid extract was subsequently used to maintain consistency with EM monolayer studies. Lipids dissolved in a mixture of chloroform and methanol (1:1) were dried under N$_2$ flux, followed by overnight incubation in a vacuum oven at 37 °C. Afterwards, lipids were fully rehydrated with buffer containing 25 mM HEPES-NaOH pH 7.4 for 10 min at room temperature, obtaining a 0.8 mg ml$^{-1}$ lipid solution. The lipid suspension was vortexed for 30 s and freeze–thawed five times in liquid nitrogen and a water bath. Mica-SLBs were then prepared by depositing LUVs onto freshly cleaved mica placed on the imaging chamber and incubated for 20 min at 37 °C with 10 mM HEPES-NaOH pH 7.4, 10 mM CaCl$_2$, and 2 mM MgCl$_2$. Samples were rinsed thoroughly with buffer containing 25 mM HEPES-NaOH pH 7.4 with 150 mM or 0–10 mM NaCl for samples in high-salt (for rings$_{HS}$) or low-salt conditions, respectively. A final volume of 300 µl of these buffers for their respective conditions was added into the imaging chamber.

### FLM image and movie acquisition

For the FLM studies, Vipp1 was chemically labeled at the N terminus with Alexa Fluor 488 TFP ester (Thermo Fisher Scientific), following the labeling procedure provided by the manufacturer. Fluorescence image acquisition was performed using an inverted spinning disc microscope assembled by 3i (Intelligent Imaging Innovation), consisting of a Nikon base (Eclipse C1, Nikon), a ×100/1.49 numerical aperture oil immersion objective, and an EVOLVE EM-CCD camera (Roper Scientific). The plugin Turboreg[63] and a custom-written ImageJ macro were used for *X*–*Y* drift correction.

### Fast scan AFM image acquisition

A JPK NanoWizard Ultraspeed AFM (Bruker and JPK BioAFM) equipped with USC-F0.3-k0.3-10 cantilevers with a spring constant of 0.3 N nm$^{-1}$ and a resonance frequency of about 300 kHz (Nanoworld) was used for image acquisition. The F-AFM was operated in tapping mode, where the cantilever oscillated at a frequency proximal to 150 kHz. Here, both topographic and phase images were reported from at least three independent experiments. Initially, the SLB was imaged before selecting and imaging the area of interests (AOIs). Imaging of the AOIs was ongoing, and Vipp1, Vipp1Δα6$_{1-219}$, or Vipp1$_{F197K/L200K}$ samples were injected into the imaging chamber at 3.5–7 µM or 21 µM for Vipp1 rings$_{HS}$ in the final volume with either high- or low-salt buffer, as described above in the LUV preparation. Images were analyzed with JPKSPM Data Processing, ImageJ, and WSxM software[64]. To calculate filament growth speed, the segmented line function in Fiji was used to measure the change in filament length between video frames. Each data point is a velocity between time points derived from 11 measured filaments.

## Negative-stain electron microscopy and data collection

To visualize Vipp1, 4 μL of 6 μM sample was spotted onto plasma-cleaned carbon-coated EM grids (300-mesh, Agar Scientific) and incubated for 1 min. The samples were blotted, washed with water, and stained twice with two drops of 2 % uranyl acetate. Images were acquired on a FEI Tecnai 12 electron microscope equipped with a TVIPS 4K CMOS XF416 camera.

## Vipp1 monolayer assay and image analysis

Monolayer assays were performed following previously outlined methods[1] but with modifications in buffer composition and lipid drop size. *E. coli* total lipid extract (Avanti Polar Lipids) was used to prepare the lipid monolayers. Wells (4 mm diameter) in a custom-built Teflon block were filled with 50 μl of assay buffer (25 mM HEPES-NaOH, pH 8.4, 5 mM KCl, and 5 mM $MgCl_2$), and a 3 μl drop of 0.1 mg ml$^{-1}$ lipid dissolved in chloroform was gently applied on top and allowed to evaporate for 1 h. Carbon-coated EM grids (200-mesh, Agar Scientific) without plasma cleaning or glow discharge were placed on the lipid layer, with the carbon side facing the lipid. Then, 15 μM Vipp1 was added through a side port and mixed gently with the buffer beneath the lipid layer. Control wells without protein or with only protein (chloroform drop without lipid) were included. The assays were incubated for 1 h, and grids were subsequently removed, stained, and imaged on a Philips Tecnai 12 electron microscope. Datasets containing 136 and 139 images for Vipp1 and Vipp1Δα6$_{1-219}$, respectively, were collected at 2.563 Å pixel size. The contrast transfer function (CTF) was estimated with CTFFIND-4.1 (ref. [65]). Filament segments were picked and extracted for three rounds of 2D classification in Relion4 (ref. [66]). The final 2D class averages presented incorporated 1,951 and 6,902 segments for Vipp1 and Vipp1Δα6$_{1-219}$, respectively.

## Cryo-EM sample preparation and data collection

Three and a half microliters of Vipp1 at a final concentration of 60 μM was incubated for 90 s on a plasma-cleaned holey R2/2 Quantifoil copper grid (Electron Microscopy Science) before plunge freezing in liquid ethane using a Vitrobot Mark IV (FEI) set at 100% humidity and 10 °C. Cryo-EM data for Vipp1$_{L1}$ were collected at 300 kV on a Titan Krios (Diamond Light Source, UK) equipped with a Gatan Quantum K2 Summit detector operated in super-resolution mode with a pixel size of 0.536 Å. A total of 19,740 videos were acquired at a defocus range of 0.75–2.5 μm, with a total electron dose of 50 e$^-$/Å$^2$ fractionated over 50 frames and an exposure of 4.1 s. For Vipp1$_{F197K/L200K}$ and Vipp1Δα6$_{1-219}$, cryo-EM data were collected at 300 kV on a Titan Krios microscope (LonCEM, The Francis Crick Institute, UK) equipped with a Gatan K3 detector operated in super-resolution mode with a pixel size of 0.55 Å. A total of 30,216 and 29,562 micrographs were acquired at a defocus range between 0.75 and 2.5 μm with a total electron dose of 50 and 41 e$^-$/Å$^2$ fractionated over 44 and 28 frames using 4.6- and 3-s exposures, respectively.

## Cryo-EM image processing and helical reconstruction of Vipp1 filaments

Micrograph videos were corrected for beam-induced sample motion and Fourier cropped to a pixel size of 1.072 Å for Vipp1 or 1.1 Å for Vipp1$_{F197K/L200K}$, and Vipp1Δα6$_{1-219}$ using MotionCor2 (ref. [67]). CTF estimations were performed using CTFFIND-4.1. Filaments were boxed into overlapping particles using crYOLO[68]. Each particle box overlapped with its neighbor by 46 Å (Vipp1$_{L1}$, Vipp1$_{F197K/L200K\_L1}$, Vipp1Δα6$_{L2}$) or 44 Å (Vipp1Δα6$_{L3}$), equivalent to the left-handed 5-start or 4-start helical pitches, respectively. Using Relion4, this resulted in extraction of 138,562, 1,172,337, and 648,059 particles with a box size of 504 pixels for Vipp1, Vipp1$_{F197K/L200K}$, and Vipp1Δα6$_{1-219}$, respectively. Particles were then binned by a factor of three and imported into cryoSPARCv3 (ref. [69]) for iterative rounds of 2D classification. Low-quality particles were discarded, and remaining particles were sorted into separate bins on the basis of the Fourier transform. Attempts to determine

helical symmetry parameters on the basis of $C_1$ reconstructions yielded only low-quality reconstructions without meaningful symmetries. Only Vipp1$_{Δα6\_L3}$ symmetry particle class averages yielded a Fourier transform with non-overlapping layer lines amenable to indexing (Extended Data Fig. 4). This produced a grid of possible symmetries that were systematically tested using helical refinement in Cryosparc. The helical parameters that yielded a reconstruction showing obvious secondary structure features were used for next steps in Relion4. This reconstruction also served as a subsequent initial reference map. For all other symmetries, including those for Vipp1$_{L1}$, Vipp1$_{F197K/L200K\_L1}$, and Vipp1$_{Δα6\_L2}$, particle class averages yielded Fourier transforms with overlapping layer lines that impeded indexing. Initial helical parameters were therefore determined by calculating and screening a grid of theoretical lattices close to Vipp1$_{Δα6\_L3}$ symmetry. Cleaned stacks containing 64936, 508,377, 91,126, and 57,329 particles relating to Vipp1$_{L1}$, Vipp1$_{F197K/L200K\_L1}$, Vipp1$_{Δα6\_L2}$, and Vipp1$_{Δα6\_L3}$ were exported from Cryosparc and re-extracted in Relion using a 150-pixel box size so that particles remained binned by a factor of three. Iterative rounds of 3D classification were undertaken resulting in final particle stacks of 43,480, 36,652, 38,361, and 38,585 relating to Vipp1$_{L1}$, Vipp1$_{F197K/L200K\_L1}$, Vipp1$_{Δα6\_L2}$, and Vipp1$_{Δα6\_L3}$, respectively. Particles were re-extracted with no binning using a 450-pixel box size for iterative rounds of 3D autorefinement incorporating three rounds of CTF refinement and Bayesian polishing. For these steps, a mask covering the central 30% of the map was used. Final refinements for Vipp1$_{L1}$, Vipp1$_{F197K/L200K\_L1}$, Vipp1$_{Δα6\_L2}$, and Vipp1$_{Δα6\_L3}$ converged with helical rises of 2.372, 2.440, 2.155, and 2.159 Å, and helical twists of −75.860°, −75.835°, 68.507°, and 85.495°, respectively. Final resolutions of 3.7, 3.7, 3.8, and 3.7 Å were based on a gold-standard Fourier shell correlation of 0.143. Vipp1$_{L1}$, Vipp1$_{Δα6\_L2}$, and Vipp1$_{Δα6\_L3}$ maps were sharpened using Phenix 1.2 (ref. [70]), whereas Vipp1$_{F197K/L200K\_L1}$ was sharpened using Relion4 postprocess. Map local resolution was estimated using ResMap[71].

## Model building and refinement

A monomer extracted from a Vipp1 $C_{14}$-symmetry ring (PDB code 6ZW4) was fitted into Vipp1$_{Δα6\_L3}$ as an initial build template. The high-quality map, with generally excellent side chain detail, facilitated manual building and modeling in Coot[72] and ISOLDE[73]. By applying the helical parameters to this asymmetric unit, a helical filament was generated in ChimeraX 1.4 (ref. [74]). A central subunit was chosen, and all subunits that were not within 5 Å of this central subunit were deleted. This assembly comprising 19 subunits was then used for model building with the central subunit modeled in the context of its neighbors. At the end of this iteration, the central subunit was extracted and a new helical filament and 19-subunit assembly were generated. This 19-subunit was used for real-space refinement in Phenix after which a new 19-subunit assembly was generated again using the central subunit. This model building process was iterated with the same workflow for Vipp1$_{L1}$, Vipp1$_{F197K/L200K\_L1}$, and Vipp1$_{Δα6\_L2}$, although a monomer from the Vipp1$_{Δα6\_L3}$ model was used as an initial build template. In the lower-resolution regions of Vipp1$_{L1}$ and Vipp1$_{Δα6\_L2}$ maps, where side chain detail was reduced or absent, side chains were modeled on the basis of Vipp1$_{Δα6\_L3}$ and Vipp1$_{F197K/L200K\_L1}$ structures. Final refinement and model validation statistics are provided in Table 1. Atomic coordinate files include matrices for the biological assembly. To generate the helical filament in Chimera, use the command sym #N or in ChimeraX use sym #N biomt, where *N* is the model number.

## Cryo-ET sample preparation, data collection, and image processing

Gold fiducials (5 nm) without BSA (Sigma-Aldrich) were mixed with 60 μM Vipp1Δα6$_{1-219}$ sample; 3.5 μL of this mix was incubated on plasma-cleaned holey R2/2 Quantifoil copper 200 mesh grids for 90 s at 10 °C and vitrified as for cryo-EM samples. Tomograms were collected at 300 kV on a Titan Krios electron microscope (LonCEM) equipped

with a Gatan K3 detector with pixel size of 2.257 Å. Dose-symmetric tilt-series were acquired from −60° to +60° with 3° intervals and a defocus range between 3 and 6 μm. A total accumulated dose of 102 e⁻/Å² was fractionated over 15 frames per tilt using an exposure of 5.6 s. Tilt-series movies were corrected for beam-induced sample motion using MotionCor2 (ref. 67). Using IMOD 4.11 (ref. 75), the tilt-series was 5-binned to yield a pixel size of 11.3 Å, aligned, and reconstructed using a SIRT algorithm. Data collection parameters are provided in Table 2.

## Reporting summary
Further information on research design is available in the Nature Portfolio Reporting Summary linked to this article.

## Data availability
3D cryo-EM density maps produced in this study have been deposited in the Electron Microscopy Data Bank under accession codes EMD-18318, EMD-18319, EMD-18321, and EMD-18322 for Vipp1$_{L1}$, Vipp1$_{F197K/L200K\_L1}$, Vipp1$_{\Delta\alpha6\_L2}$, and Vipp1$_{\Delta\alpha6\_L3}$, respectively. Atomic coordinates have been deposited in the Protein Data Bank (PDB) under PDB IDs 8QBR, 8QBS, 8QBV, and 8QBW, respectively. Source data are available with the manuscript online. AFM data are available at https://zenodo.org/records/13149421 (ref. 76). Source data are provided with this paper.

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

## Acknowledgements
Cryo-EM data were collected at Diamond Light Source and London Consortium for CryoEM (LonCEM). We thank N. Cronin (LonCEM, The Francis Crick Institute) for cryo-EM data collection support, P. Simpson for in-house EM support, and S. Islam for in-house computational support. We thank Diamond for access to and support for the cryo-EM facilities at the UK national electron bio-imaging center (eBIC) funded by the Wellcome Trust, MRC, and BBSRC. J.S. was funded by a BBSRC grant (BB/W008181/1) to H.H.L. This work was funded by a Wellcome Trust Senior Research Fellowship (215553/Z/19/Z) to H.H.L. A.R. acknowledges funding from the Swiss National Fund for Research, grants no. 310030_200793/1 and no. CRSII5_189996, and from the European Research Council, Synergy Grant N°951324_R2-TENSION. J.E. acknowledges support from EMBO long-term post-doctoral fellowship ALTF 989-2022. A.C. acknowledges funding from MCIU, PID2022-140687NB-I00; MCIU/AEI/FEDER MINECOG19/P66, RYC2018-024686-I, and Basque Government IT1625-22. A.M. is a recipient of a predoctoral fellowship from the Basque Government.

## Author contributions
J.E. and A.R. conceived the in vitro assays for photonic microscopy. J.E. performed all photonic microscopy experiments. A.M. and A.C. designed and conceived Fast-AFM experiments. A.M. and A.C. gathered and analyzed Fast-AFM data. S.N. and H.H.L. designed and conceived the EM studies. S.N. purified Vipp1 samples and undertook NS EM and cryo-EM studies. S.N. and H.H.L. collected and processed cryo-EM data and built helical structures. J.S. contributed to Vipp1 purification. A.C. and H.H.L. analyzed the data and wrote the paper with contributions and revisions from all authors.

## Competing interests
The authors declare no competing interests.

## Additional information
**Extended data** is available for this paper at https://doi.org/10.1038/s41594-024-01401-8.

**Correspondence and requests for materials** should be addressed to Adai Colom or Harry H. Low.

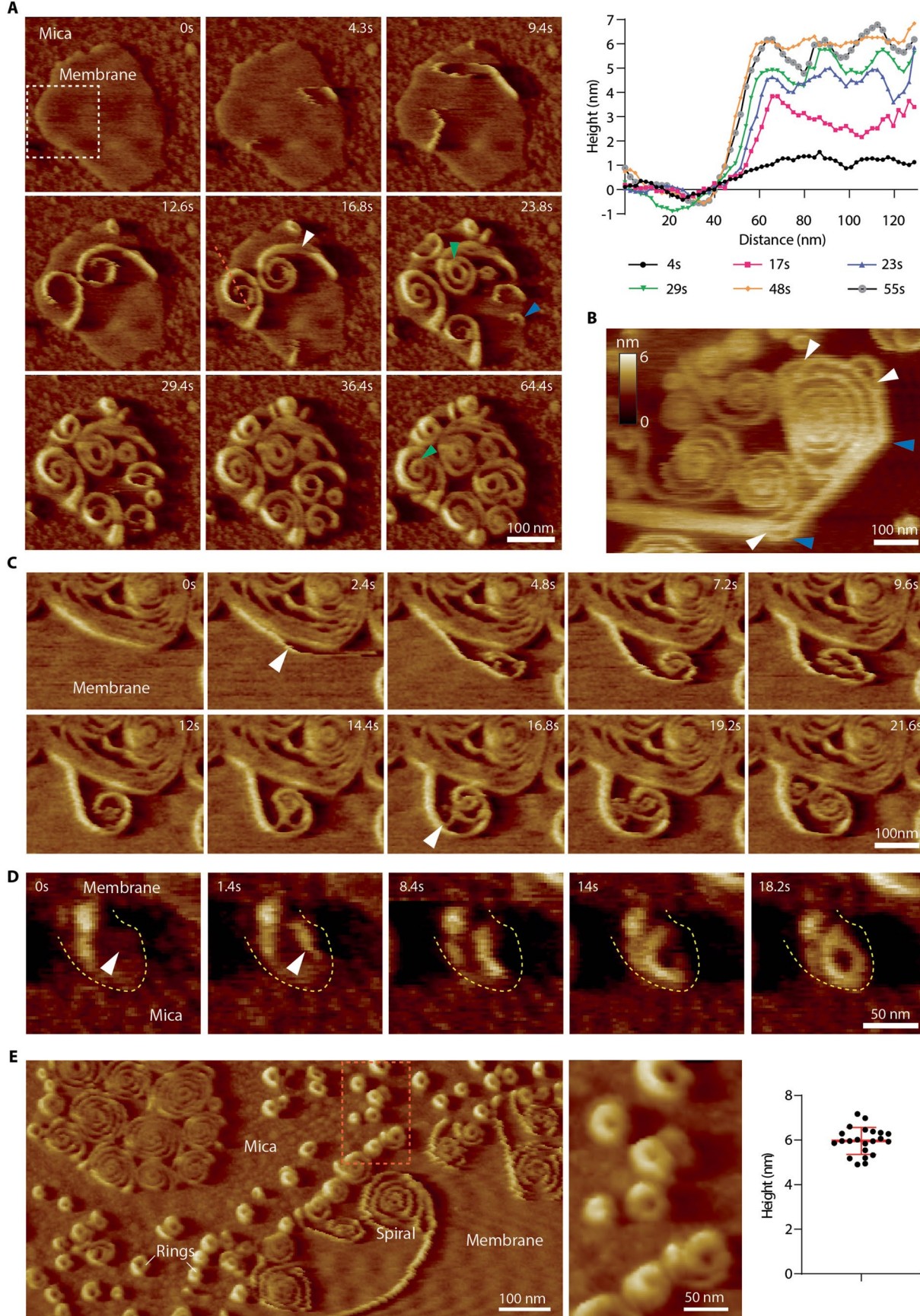

**Extended Data Fig. 1 | See next page for caption.**

**Extended Data Fig. 1 | Vipp1 assembles dynamic networks of sheets, spirals and rings on membrane, related to Figs. 2 and 3. a**, Phase F-AFM timecourse showing spiral formation and ring$_{LS}$ biogenesis. Highlighted events include ring$_{LS}$ abscission, filament branching, and ring formation in the absence of a spiral (green, white and blue arrows, respectively). White dotted box indicates zoomed region relating to Fig. 3a. Red dotted line relates to plotted height profile (right). Scan rate 150 Hz, 256 x 213 pixels. **b**, Average F-AFM height image showing the outer turn of a spiral formed by linear sheet connected by vertices. Here, lattice breaks in the filament packing marked shifts in polymerisation direction (blue arrows). White arrows indicate sheet branching. Scan rate 35 Hz, 256 x 151 pixels. **c**, Phase F-AFM timecourse showing how filaments often grow alongside established filaments with a propensity to branch off and form spirals (white arrows). Scan rate 42 Hz, 256 x 179 pixels. **d**, Phase F-AFM timecourse showing ring formation due to spatial constraint (white arrow). Dotted yellow line delineates membrane boundary. Scan rate 150 Hz, 256 x 213 pixels. **e**, Phase F-AFM image showing examples of rings formed in the absence of a spiral. These rings form on membrane micro-patches on the mica due to spatial constraint. Scan rate 15 Hz, 512 x 384 pixels. Middle panel relates to dotted box in left panel. The plot (right) shows the rings with mean height 6.0 ± 0.6 nm. Data derived from N = 21 independent measurements. Error bars show one standard deviation of the mean.

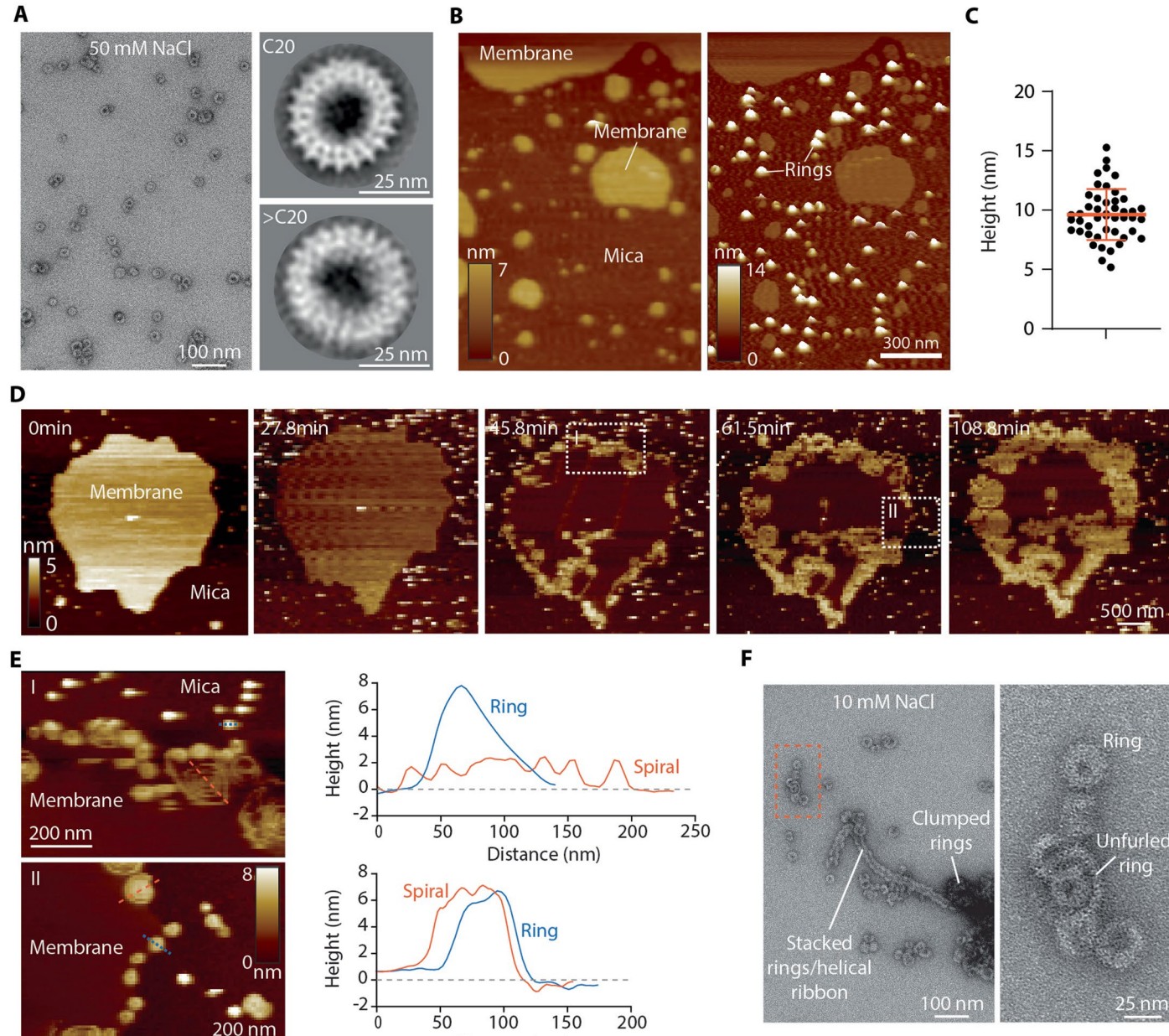

**Extended Data Fig. 2 | Vipp1 rings_HS form planar spirals in low salt conditions, related to Fig. 3. a**, Vipp1 rings_HS purified in high salt (50 mM NaCl) form rings between C11-C17 symmetries[1] and also with higher symmetries such as C20 or above with diameters of 41 nm and 43 nm, respectively. Experiment repeated independently N = 3. **b**, Vipp1 rings_HS bind to mica as stable pre-formed dome-shaped rings[1] in a high salt (150 mM NaCl) buffer. Scan rate 8 Hz, 256 x 146 pixels. Left panel represents field of view before sample addition. Right panel is complementary to Fig. 3h. **c**, Plot showing Vipp1 rings_HS heights as measured by F-AFM. Mean height is lower than Vipp1 rings_HS heights determined by cryo-EM between ~15-21 nm for C11-C17 symmetries[1]. Data derived from N = 46 independent measurements. Error bars show one standard deviation of the mean. **d**, F-AFM timecourse showing how, in a low salt (10 mM NaCl) buffer, Vipp1 rings_HS forms planar sheets and spiral networks on membrane. Scan rate 20 Hz, 256 x 204 pixels. **e**, F-AFM image highlighting dotted box regions I and II in **D**. Rings and spirals nucleate and assemble on the highly curved membrane edge. Blue and red dotted lines were used for plotted height profiles (right) of typical spiral and rings. Scan rate 30 Hz, 256 x 179 pixels. **f**, NS EM image of Vipp1 rings_HS after buffer exchange into 10 mM NaCl buffer. Compared with the sample purified in 50 mM NaCl as in **A**, rings tend to be clumped or stacked and in the process of unfurling. Dotted box indicates zoomed region (right panel). Experiment repeated independently N = 3.

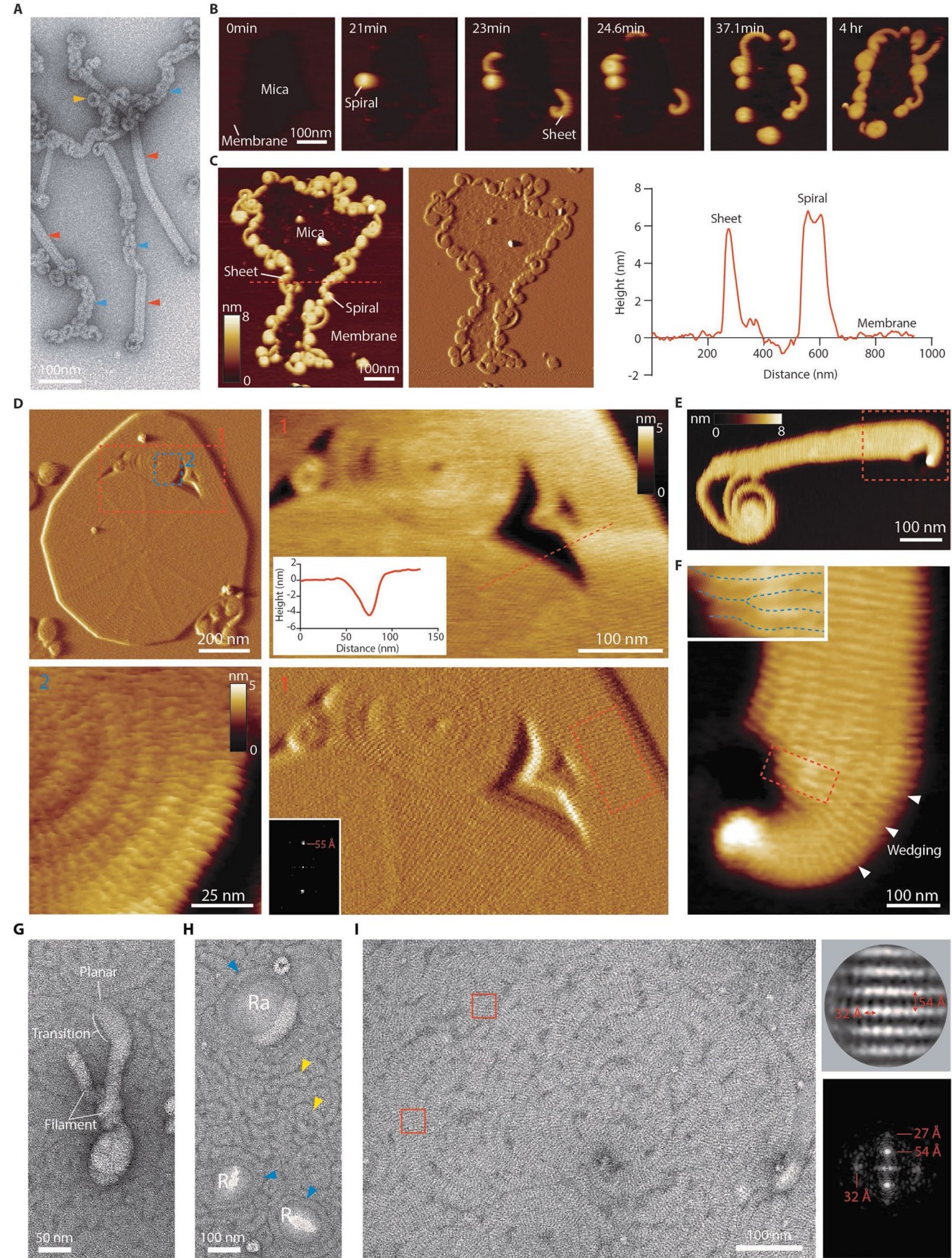

**Extended Data Fig. 3 | See next page for caption.**

**Extended Data Fig. 3 | Vipp1Δα6$_{1-219}$ forms tightly packed planar spirals and highly ordered sheets, related to Figs. 2–4. a**, NS EM image showing Vipp1Δα6$_{1-219}$ forming helical filaments, helical-like ribbons, and rings (red, blue and yellow arrows, respectively). Related to Fig. 5a. **b**, F-AFM phase timecourse showing Vipp1Δα6$_{1-219}$ recruitment to the membrane where it forms curled sheets and compact spirals on the membrane edge. Scan rate 20 Hz, 256 x 213 pixels. **c**, F-AFM height and phase image showing membrane edges decorated with sheet and compact spirals. Red dotted line indicates plotted height profiles (right) with Vipp1Δα6$_{1-219}$ planar filaments/sheet equivalent height to native Vipp1, related to Fig. 2e. Scan rate 5 Hz, 256 x 320 pixels. **d**, Vipp1Δα6$_{1-219}$ forms dense crystalline planar sheets and spirals. Scan rate 10 Hz, 512 x 426 pixels. Dotted red line in top right panel was used for the plotted height profile (inset). Scan rate 10 Hz, 256 x 154 pixels. Dotted box in bottom right panel indicates region used for Fourier Transform (inset). Parallel ridges are separated by a 55 Å spacing. **e**, F-AFM image showing Vipp1Δα6$_{1-219}$ planar sheet curling at one end and branching into a spiral at the other. Scan rate 10 Hz, 256 x 488 pixels. **f**, Zoom of dotted box in **E**, showing how planar sheets curl via a wedging mechanism on the outer side of the bend (white arrows). Concurrently, sections are removed from the inner side (red dotted box and inset). Scan rate 10 Hz, 128 x 160 pixels. **g**, NS EM image showing Vipp1Δα6$_{1-219}$ rod-like filaments decorating the surface of a monolayer which is itself covered by 2D planar filaments. Experiment repeated independently N = 3. **h**, Vipp1Δα6$_{1-219}$ forms a mosaic of 2D planar filaments (blue arrows) and rings (yellow arrows) on a lipid monolayer. Filaments encircle apparent raised areas (Ra) and regions where the monolayer is ruptured (R). Experiment repeated independently N = 3. **i**, Zoom view of Vipp1Δα6$_{1-219}$ mosaic of 2D planar filaments and sheets on a lipid monolayer, related to Fig. 4d. Note spiral filaments are not observed here. Red boxes indicate example regions for particle extraction and alignment. Top right shows particle class average and corresponding Fourier Transform (bottom right). Filament stripes are 54 Å apart with an orthogonal 32 Å repeat.

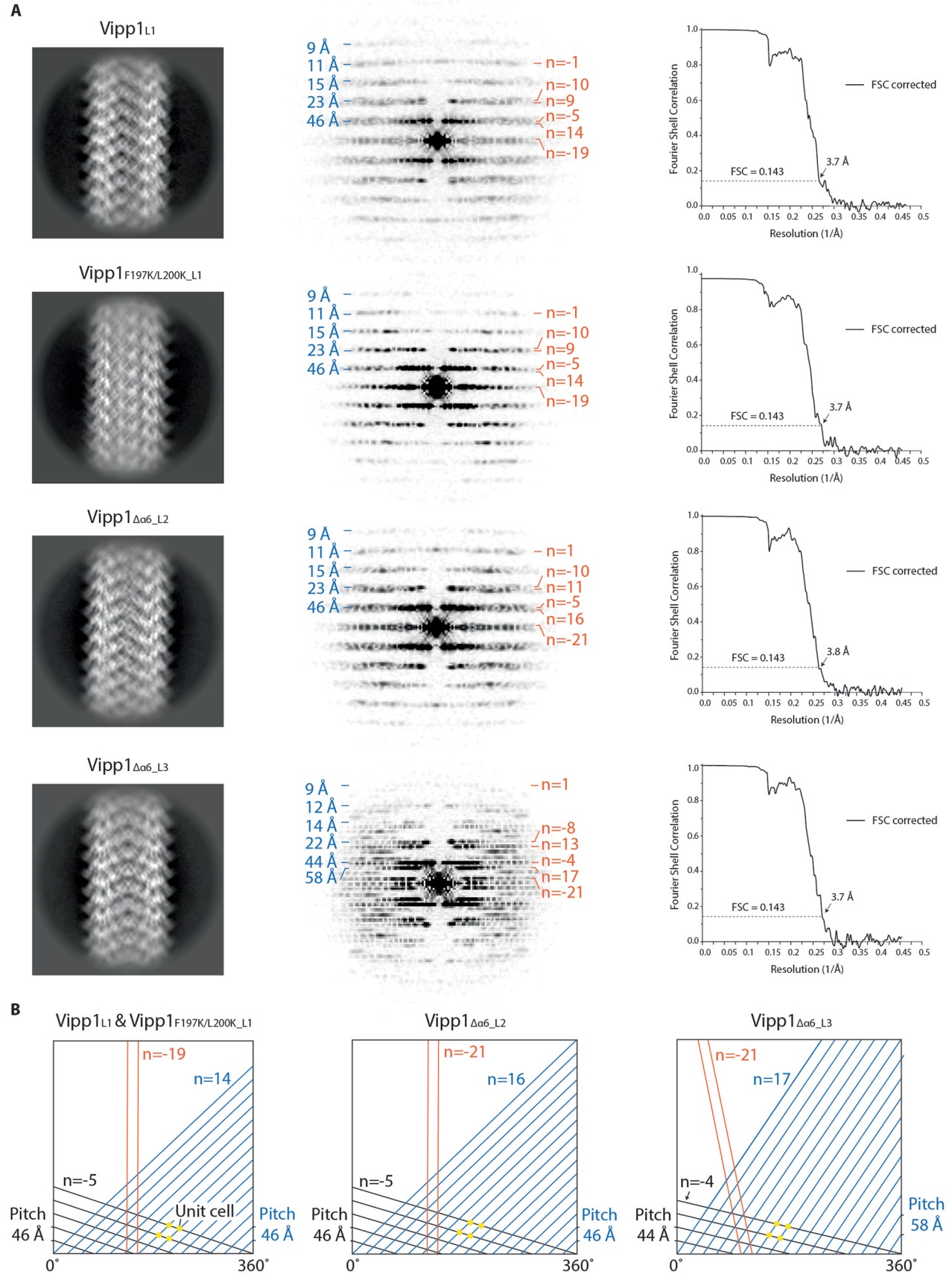

**Extended Data Fig. 4 | Vipp1 helical lattice analyses and map resolutions.**
**a**, 2D class averages of four different helical filaments including Vipp1$_{L1}$, Vipp1$_{F197K/L200K\_L1}$, Vipp1$_{\Delta\alpha6\_L2}$ and Vipp1$_{\Delta\alpha6\_L3}$ (left column) with Fourier Transforms and Bessel Orders for key layer lines assigned (middle columns). Associated gold standard FSC curves for each map are presented (right column). **b**, Vipp1$_{L1}$, Vipp1$_{F197K/L200K\_L1}$, Vipp1$_{\Delta\alpha6\_L2}$ and Vipp1$_{\Delta\alpha6\_L3}$ helical assemblies drawn as 2D lattices showing arrangement of key Bessel orders.

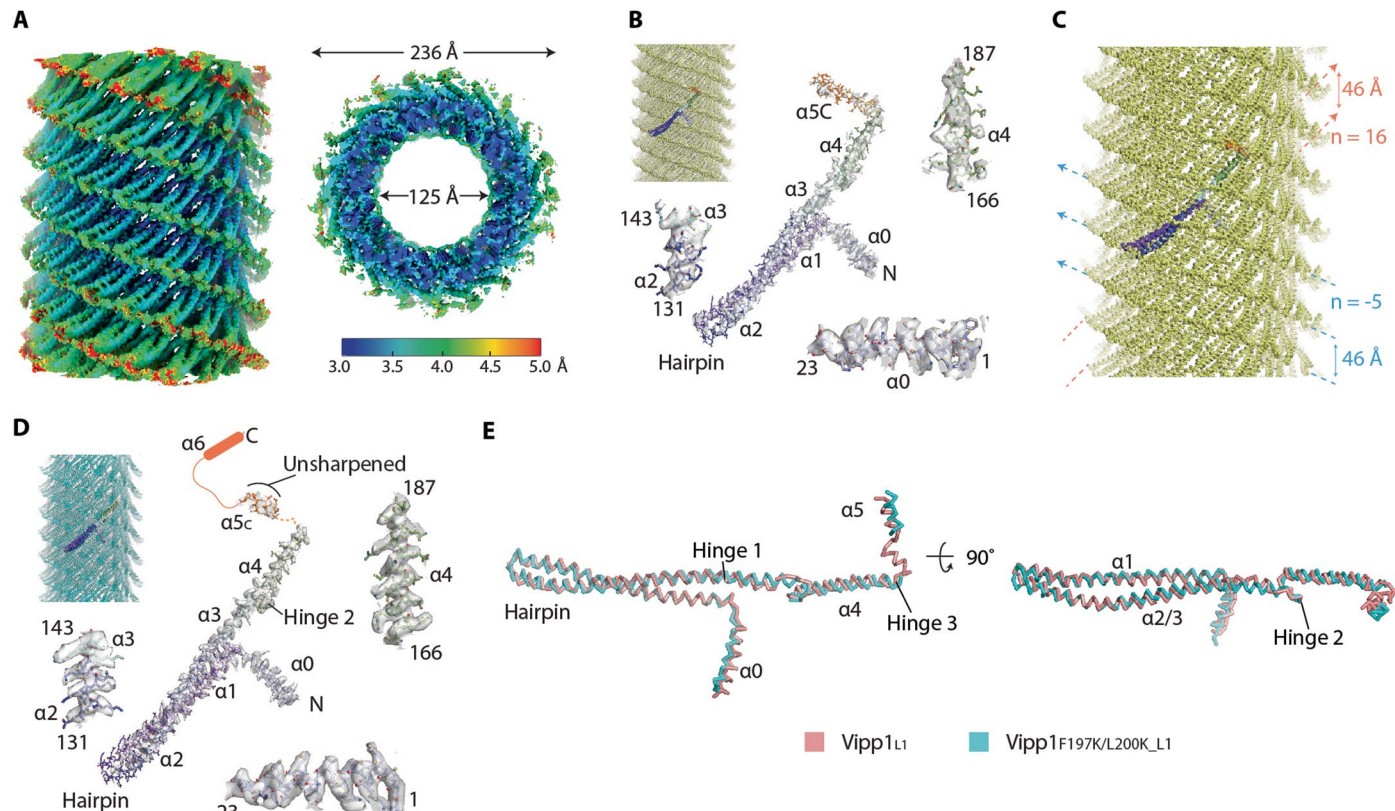

**Extended Data Fig. 5 | Cryo-EM maps and model build for Vipp1$_{\Delta\alpha6\_L2}$ and Vipp1$_{F197K/L200K\_L1}$, related to Figs. 5 and 6. a**, Sharpened Vipp1$_{\Delta\alpha6\_L2}$ map contoured at 2σ showing local resolution estimates. Vipp1$_{\Delta\alpha6\_L2}$ map was resolved to 3.8 Å overall using helical parameters with rise = 2.16 Å and rotation = 68.50°. **b**, Vipp1$_{\Delta\alpha6\_L2}$ map fitted with Vipp1$_{\Delta\alpha6\_L2}$ helical filament structure (top left). The coloured monomer is isolated and zoomed to show map quality, build and fit. Map contoured between 3-4σ. Map local resolution range was broader than for Vipp1$_{\Delta\alpha6\_L3}$ with lower map quality particularly around helix α5. The Vipp1$_{\Delta\alpha6\_L2}$ lattice therefore has increased instability in comparison to Vipp1$_{\Delta\alpha6\_L3}$ although residual heterogeneity in the particle stack cannot be discounted as a contributory factor in reducing map quality. **c**, Structure of the Vipp1$_{\Delta\alpha6\_L2}$ helical filament. Bessel orders n = 16 and n = −5 are indicated with pitch. Specifically, the ESCRT-III-like protofilaments form a 16-start right-handed helix which twist around the helical axis with a 46 Å pitch. Concurrently, these protofilaments

align to form a 5-start left-handed helix with a 46 Å pitch. In comparison to Vipp1$_{\Delta\alpha6\_L3}$ (Fig. 5 and Extended Data Fig. 4), the effect is to rotate the Vipp1 lattice −13° clockwise around the helical axis with modest packing adjustments accommodating a minor flattening of the rhomboid unit cell. This results in a slightly narrower filament 23.6 nm in diameter with a hollow inner lumen diameter of 12.5 nm. Overall, the Vipp1$_{\Delta\alpha6\_L2}$ and Vipp1$_{\Delta\alpha6\_L3}$ structures reveal how subtle shifts in Vipp1 assembly and lattice packing modulate inherent polymer stability. **d**, Vipp1$_{F197K/L200K\_L1}$ map fitted with Vipp1$_{F197K/L200K\_L1}$ helical filament structure (top left). The coloured monomer is isolated and zoomed to show map quality, build and fit. Map contoured at 4σ. Related to Fig. 6. **e**, Superposition of subunits from Vipp1$_{F197K/L200K\_L1}$ and Vipp1$_{L1}$ with RMSD Cα = 1.8 Å. Subunits share similar conformations although Vipp1$_{F197K/L200K\_L1}$ does not have the hairpin kink observed in Vipp1$_{L1}$ (Extended Data Fig. 6D).

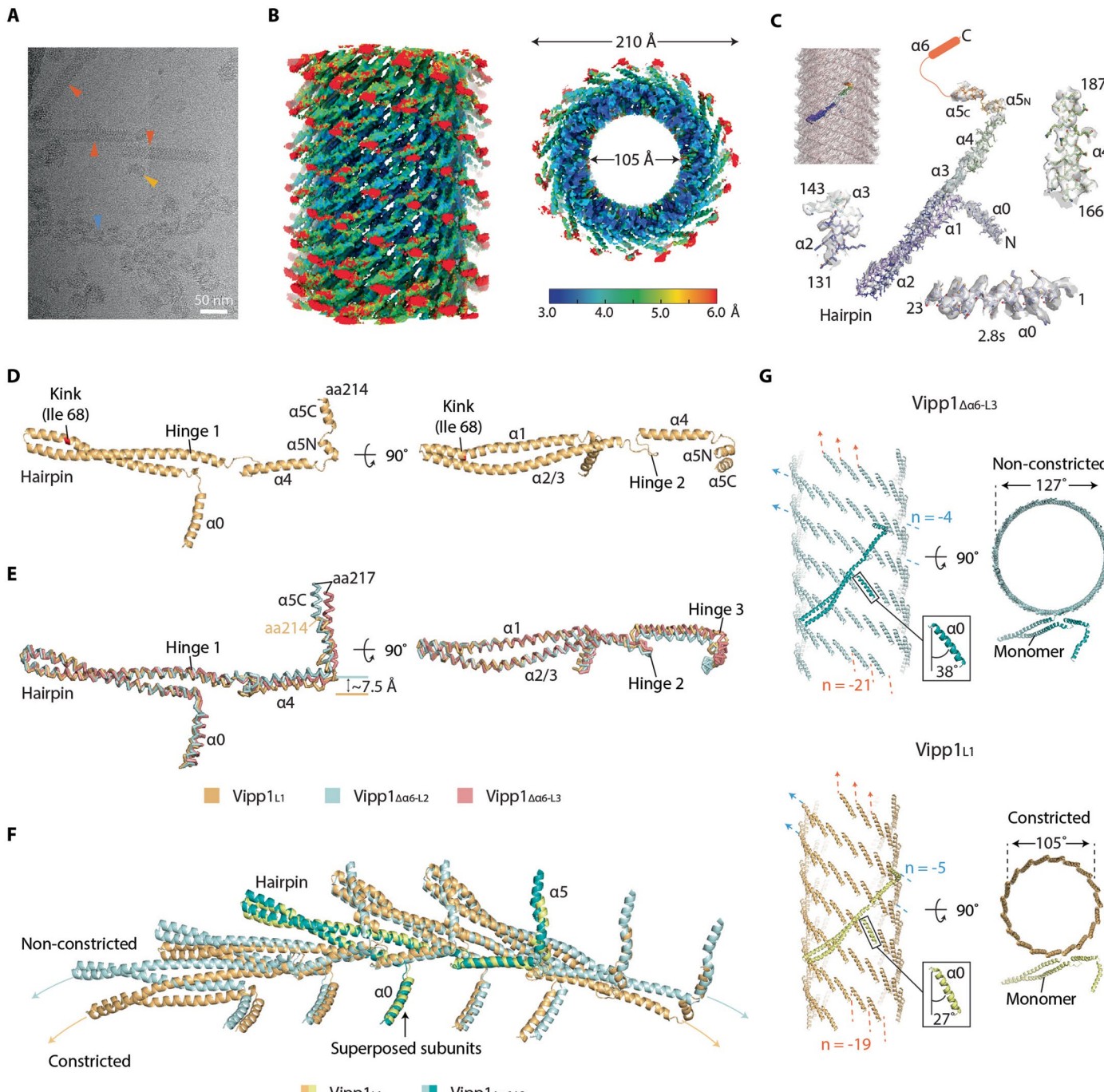

**Extended Data Fig. 6 | Comparison of Vipp1$_{L1}$ with Vipp1$_{\Delta\alpha6\_L3}$ yields a mechanism of filament constriction. a**, Cryo-EM image showing Vipp1$_{L1}$ forming helical filaments, helical-like ribbons, and rings (red, blue and yellow arrows, respectively). A zoomed version is presented as Supplementary Fig. 1b. **b**, DeepEMhancer[77] post-processed Vipp1$_{L1}$ map contoured at 3σ showing local resolution estimates. **c**, Vipp1$_{L1}$ map fitted with Vipp1$_{L1}$ helical filament structure (top left). The coloured monomer is isolated and zoomed to show map quality, build and fit. Map contoured between 2.8-3.5σ. The map region relating to helix α5c, contoured at 3σ, was not DeepEMhancer processed or sharpened. **d**, Structure of a Vipp1$_{L1}$ subunit. **e**, Superposition of Vipp1$_{L1}$, Vipp1$_{\Delta\alpha6\_L2}$ and

Vipp1$_{\Delta\alpha6\_L3}$ monomers with the alignment focussed on the hairpin motif. The Cα RMSD of Vipp1$_{L1}$ subunits to Vipp1$_{\Delta\alpha6\_L2}$ or Vipp1$_{\Delta\alpha6\_L3}$ was 1.5 Å and 1.7 Å, respectively. Note how helix α5 is compacted in Vipp1$_{L1}$ compared with Vipp1$_{\Delta\alpha6\_L2}$ and Vipp1$_{\Delta\alpha6\_L3}$. **f**, Comparison of constricted versus non-constricted ESCRT-III-like protofilaments in Vipp1$_{L1}$ and Vipp1$_{\Delta\alpha6\_L3}$. Five monomers were superposed with the alignment focussed on the central subunit. **g**, Helical constriction occurs through subunit loss from the circumferential belt. Vipp1$_{\Delta\alpha6\_L3}$ and Vipp1$_{L1}$ filaments are compared, with constriction occurring due to a loss of two subunits (n = −21 reduced to n = −19) coupled with ~11 Å rotation of the left-handed four-start helix as indicated in the helix α0 zoom boxes.

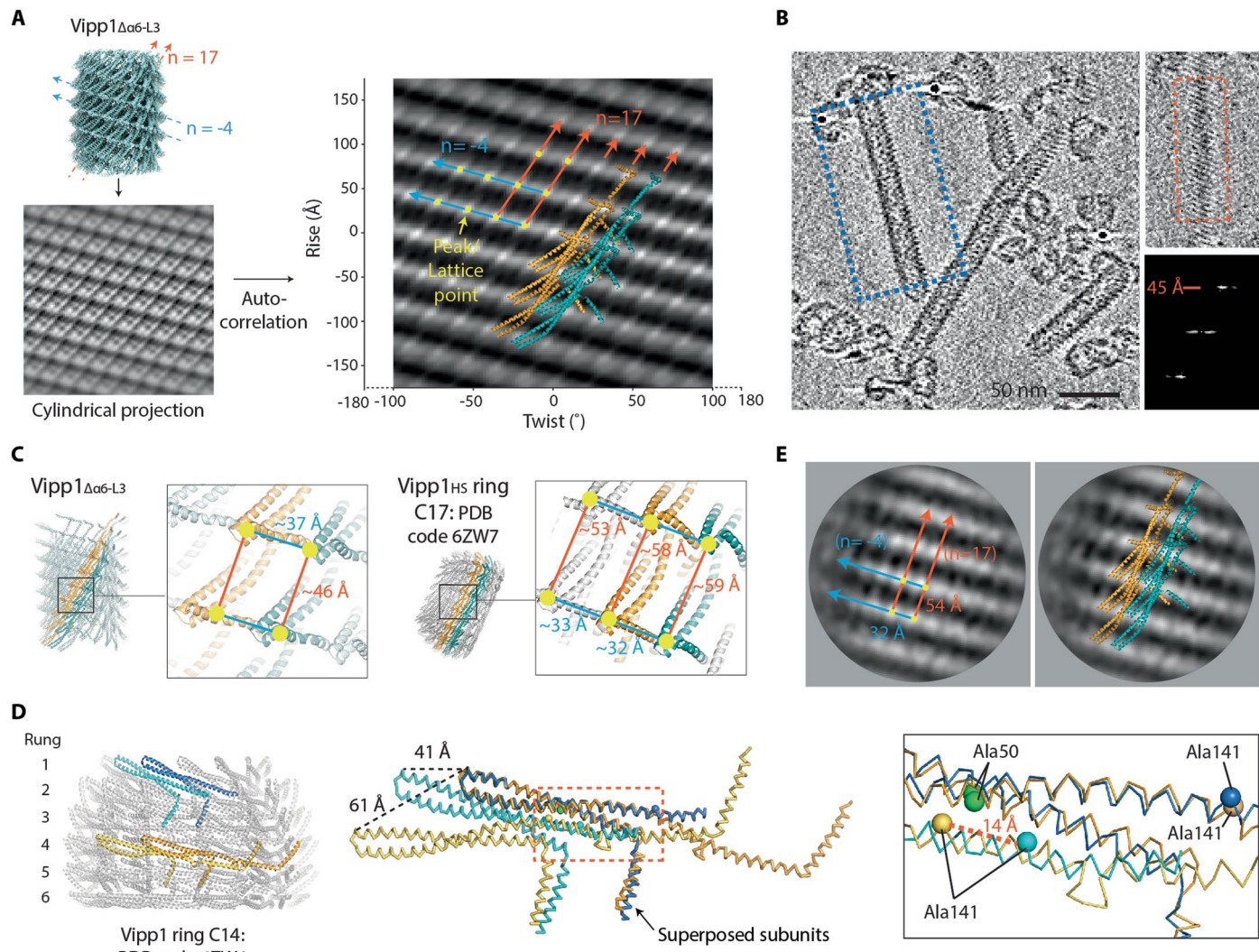

**Extended Data Fig. 7 | Analysis of Vipp1 planar sheet and spiral ultrastructure. a**, Cylindrical projection of the Vipp1$_{\Delta\alpha6\_L3}$ map with its auto-correlation function[56]. Based on the Vipp1$_{\Delta\alpha6\_L3}$ structure, the orientation of six subunits have been superposed on the 2D lattice with the ESCRT-III-like protofilaments coloured orange and turquoise. **b**, Cryogenic electron tomogram of Vipp1$_{\Delta\alpha6\_L3}$. Central slice through the helical filaments is shown (left panel). Top right panel shows blue dotted box region from left panel. Only the upper surface ridges or stripes of the helical filament are sliced here. Bottom right panel shows Fourier Transform of red dotted box region. The ridges have a 45 Å repeat, which is consistent with the Vipp1$_{\Delta\alpha6\_L3}$ left-handed 4-start helix pitch. **c**, Inter-subunit packing measurements of Vipp1$_{\Delta\alpha6\_L3}$ and the equivalent in selected rungs of the Vipp1$_{HS}$ C17 symmetry ring[1]. **d**, Hairpins slide relative to each other in Vipp1 ESCRT-III-like protofilaments. Two neighbouring subunits

from rungs 1 and 4 of Vipp1$_{HS}$ C14 ring are superposed using the right-hand subunits only (left and middle panels). Whilst Ala50 and Ala 141 are aligned in the superposed subunits (dark orange and blue), Ala141 is up to 14 Å apart between the neighbouring subunits (light orange and cyan; right panel) indicating hairpin sliding. The distances 41 Å and 61 Å (middle panel) relate to the typical span of minimum and maximum inter-ridge distances observed in Vipp1 rings$_{HS}$, respectively. **e**, Lattice and subunit assignment of Vipp1Δα6$_{1-219}$ 2D planar filaments and sheets on a lipid monolayer, relating to Extended Data Fig. 3I. The class average (left) is shown with lattice points and dimensions derived from the Fourier Transform shown in Extended Data Fig. 3I. The same class average (right) is shown with Vipp1 ESCRT-III-like protofilaments equivalent to Bessel order n = 17 in Vipp1$_{\Delta\alpha6\_L3}$ positioned based on the lattice points (orange and turquoise).

# Reporting Summary

## Statistics

For all statistical analyses, confirm that the following items are present in the figure legend, table legend, main text, or Methods section.

| n/a | Confirmed | |
|---|---|---|
| ☐ | ☒ | The exact sample size (*n*) for each experimental group/condition, given as a discrete number and unit of measurement |
| ☐ | ☒ | A statement on whether measurements were taken from distinct samples or whether the same sample was measured repeatedly |
| ☒ | ☐ | The statistical test(s) used AND whether they are one- or two-sided *Only common tests should be described solely by name; describe more complex techniques in the Methods section.* |
| ☒ | ☐ | A description of all covariates tested |
| ☒ | ☐ | A description of any assumptions or corrections, such as tests of normality and adjustment for multiple comparisons |
| ☐ | ☒ | A full description of the statistical parameters including central tendency (e.g. means) or other basic estimates (e.g. regression coefficient) AND variation (e.g. standard deviation) or associated estimates of uncertainty (e.g. confidence intervals) |
| ☒ | ☐ | For null hypothesis testing, the test statistic (e.g. *F*, *t*, *r*) with confidence intervals, effect sizes, degrees of freedom and *P* value noted *Give P values as exact values whenever suitable.* |
| ☒ | ☐ | For Bayesian analysis, information on the choice of priors and Markov chain Monte Carlo settings |
| ☒ | ☐ | For hierarchical and complex designs, identification of the appropriate level for tests and full reporting of outcomes |
| ☒ | ☐ | Estimates of effect sizes (e.g. Cohen's *d*, Pearson's *r*), indicating how they were calculated |

*Our web collection on statistics for biologists contains articles on many of the points above.*

## Software and code

Policy information about availability of computer code

| Data collection | Cryo-EM data: EPU 3 software. |
|---|---|
| Data analysis | Light microscopy: The plugin Turboreg63 and a custom-written ImageJ macro were used for X-Y drift correction. AFM:JPKSPM Data Processing, ImageJ, and WSxM software. Cryo-EM: Cryosparc v3, Relion4, Phenix 1.2, IMOD 4.11 (SIRT) |

For manuscripts utilizing custom algorithms or software that are central to the research but not yet described in published literature, software must be made available to editors and reviewers. We strongly encourage code deposition in a community repository (e.g. GitHub). See the Nature Portfolio guidelines for submitting code & software for further information.

## Data

Policy information about availability of data

All manuscripts must include a data availability statement. This statement should provide the following information, where applicable:
- Accession codes, unique identifiers, or web links for publicly available datasets
- A description of any restrictions on data availability
- For clinical datasets or third party data, please ensure that the statement adheres to our policy

3D cryo-EM density maps produced in this study have been deposited in the Electron Microscopy Data Bank with accession code EMD-18318, EMD-18319, EMD-18321 and EMD-18322 for Vipp1L1, Vipp1F197K/L200K_L1, Vipp1delta6_L2 and Vipp1delta6_L3, respectively. Atomic coordinates have been deposited in the Protein Data Bank (PDB) under PDB IDs 8QBR, 8QBS, 8QBV and 8QBW, respectively.

# Research involving human participants, their data, or biological material

Policy information about studies with human participants or human data. See also policy information about sex, gender (identity/presentation), and sexual orientation and race, ethnicity and racism.

| | |
|---|---|
| Reporting on sex and gender | Not applicable |
| Reporting on race, ethnicity, or other socially relevant groupings | Not applicable |
| Population characteristics | Not applicable |
| Recruitment | Not applicable |
| Ethics oversight | Not applicable |

Note that full information on the approval of the study protocol must also be provided in the manuscript.

# Field-specific reporting

Please select the one below that is the best fit for your research. If you are not sure, read the appropriate sections before making your selection.

☒ Life sciences    ☐ Behavioural & social sciences    ☐ Ecological, evolutionary & environmental sciences

For a reference copy of the document with all sections, see nature.com/documents/nr-reporting-summary-flat.pdf

# Life sciences study design

All studies must disclose on these points even when the disclosure is negative.

| | |
|---|---|
| Sample size | statistical comparisons not relevant in this study. for AFM all regions of interest were measured in collected images. the full spread of the data was then displayed graphically with mean and standard deviation. |
| Data exclusions | No data excluded |
| Replication | All experiments were repeated at least three times |
| Randomization | AFM images and timecourses were collected on randomly chosen regions of the mica where membrane support was bound |
| Blinding | blinding not relevant to this study as relates to cryo-EM structural studies and visualisation of Vipp1 filaments by AFM |

# Reporting for specific materials, systems and methods

We require information from authors about some types of materials, experimental systems and methods used in many studies. Here, indicate whether each material, system or method listed is relevant to your study. If you are not sure if a list item applies to your research, read the appropriate section before selecting a response.

## Materials & experimental systems

| n/a | Involved in the study |
|---|---|
| ☒ ☐ | Antibodies |
| ☒ ☐ | Eukaryotic cell lines |
| ☒ ☐ | Palaeontology and archaeology |
| ☒ ☐ | Animals and other organisms |
| ☒ ☐ | Clinical data |
| ☒ ☐ | Dual use research of concern |
| ☒ ☐ | Plants |

## Methods

| n/a | Involved in the study |
|---|---|
| ☒ ☐ | ChIP-seq |
| ☒ ☐ | Flow cytometry |
| ☒ ☐ | MRI-based neuroimaging |

