## [Peer Review File · Nature Structural & Molecular Biology]

Peer Review Information

Manuscript Title: Mechanism for Vipp1 spiral formation, ring biogenesis and membrane repair

Corresponding author name(s): Harry Low, Adai Colom

Reviewer Comments & Decisions:

Decision Letter, initial version:

Message: 1st Dec 2023

Dear Dr. Low,

Thank you again for submitting your manuscript "Mechanism for Vipp1 spiral formation, ring biogenesis and membrane repair". I apologize for the delay in responding, which resulted from the difficulty in obtaining suitable referee reports. Nevertheless, we now have comments (below) from the 3 reviewers who evaluated your paper. In light of those reports, we remain interested in your study and would like to see your response to the comments of the referees, in the form of a revised manuscript.

You will see that while reviewers appreciate the results, they raise several concerns which will need to be addressed in a revision. Specifically, we ask that you revise the reporting of AFM data and results in line with reviewer's #1 comments. We agree with reviewer's #2 points improving the clarity of reporting cryo-EM results, and requests to justify multiple methods of SLB preparation.

Please be sure to address/respond to all concerns of the referees in full in a point-by-point response and highlight all changes in the revised manuscript text file. If you have comments that are intended for editors only, please include those in a separate cover letter.

We expect to see your revised manuscript within 12 weeks. If you cannot send it within this time, please contact us to discuss an extension; we would still consider your revision, provided that no similar work has been accepted for publication at NSMB or published elsewhere.

Reporting Summary:

When submitting the revised version of your manuscript, please pay close attention to our [href="https://www.nature.com/nature-portfolio/editorial-policies/image-integrity">Digital Image Integrity Guidelines](https://www.nature.com/nature-portfolio/editorial-policies/image-integrity). and to the following points below:

Please note that all key data shown in the main figures as cropped gels or blots should be presented in uncropped form, with molecular weight markers. These data can be aggregated into a single supplementary figure item. While these data can be displayed in a relatively informal style, they must refer back to the relevant figures. These data should be submitted with the final revision, as source data, prior to acceptance, but you may want to start putting it together at this point.

Data availability: this journal strongly supports public availability of data. All data used in accepted papers should be available via a public data repository, or alternatively, as Supplementary Information. If data can only be shared on request, please explain why in your Data Availability Statement, and also in the correspondence with your editor. Please note that for some data types, deposition in a public repository is mandatory - more information on our data deposition policies and available repositories can be found below: <https://www.nature.com/nature-research/editorial-policies/reporting-standards#availability-of-data>

Nature Structural & Molecular Biology is committed to improving transparency in authorship. As part of our efforts in this direction, we are now requesting that all authors identified as 'corresponding author' on published papers create and link their Open Researcher and Contributor Identifier (ORCID) with their account on the Manuscript Tracking System (MTS), prior to acceptance. This applies to primary research papers only. ORCID helps the scientific community achieve unambiguous attribution of all scholarly contributions. You can create and link your ORCID from the home page of the MTS by clicking on 'Modify my Springer Nature account'. For more information please visit please visit www.springernature.com/orcid.

[Redacted]

Sincerely,
Kat

Katarzyna Ciazynska, PhD
(she/her)
Associate Editor

Nature Structural & Molecular Biology
<https://orcid.org/0000-0002-9899-2428>

Referee expertise:

Referee #1: AFM

Referee #2: cell trafficking, structural biology

Referee #3: membrane remodelling, structural biology

Reviewers' Comments:

Reviewer #1:

Remarks to the Author:

This paper uses F-AFM (usually referred to as HSAFM) and cryoEM to explore the dynamic behaviour of Vipp1 structures on lipid bilayers in exquisite detail. The major finding of this work is that the structure of VIPP1 filaments is dynamic and changes to accommodate the curvature required to form spirals, as has been observed for its homolog ESCRT. They show that these filaments can form sheets, which branch off into spiral forming filaments, and that these phases can co-exist together. They show that the sheets are formed of merged filaments, due to their regular spacing, which matches the width of the filament. They probe deeper into the spiral formation, demonstrating that at the centre of the spiral a protruding ring forms, by pinching off the spiral, and that these rings, when purified alone, preferentially bind to defects in membranes.

The paper then uses high resolution AFM (taken at lower speeds) to observe the regular crystalline structure of the filaments, and use fourier analysis to determine the crystal packing of the lattice. They show that the lattice structure is identical across the higher order phases of spirals, rings and sheets. They probe this deeper by analysing these structures with CryoEM, showing striking similarity between the AFM and CryoEM ridge spacing (54Å). They show that these can slide back and forth, with hairpins at each ridge dictating the curvature, by allowing the filaments to come closer or further apart (41-61 Å). Beyond this point, extreme curvature is induced by broken filaments, "wedging" or through tilt, which then leads to ring formation, causing the ring to protrude. This very detailed structural study is well evidenced throughout, I have listed below a number of minor points below that it would be useful for the authors to address:

AFM images do not have z scales - this is important to allow interpretation of the changing heights across the different structures

For Fig. 1F-I : add the N for the data in the caption

For the data in Fig. 1F it would be good to have an explanation (either in the text or the methods) of how the filament growth speed was calculated

Is there a reason that the N for Fig. 1G (filament width) is much lower than the other measured parameters?

The height profile in Fig. 2B is missing the x axis label

A legend is required for Fig. 2H to show which line (blue or red) corresponds to the HS or LS rings

The AFM in Fig. 3 is said to be taken at a lower scan rate - is it appropriate to continue

referring to it as F-AFM at this point, or is it now just standard tapping mode AFM?

In Fig. 3B the red dotted boxes are difficult to see over the AFM images - you may want to consider using a different colour for these.

It would be useful to make Fig. 4A larger, as currently it is hard to see the structures that are being highlighted. The same goes for Fig. 5A

In the AFM image in Fig. 6B, the numbers in red are not easily visible - consider changing the colour of this.

It would be important for the authors to add more detail to the AFM methods section.

Given that high speed AFM was important to capture the dynamics of the Vipp, it would be important to provide the scan rate used. This is also true for the high resolution imaging carried out for Fig. 3. Additionally, the resolution that the AFM was taken at should also be provided.

Reviewer #2:

Remarks to the Author:

Eukaryotic ESCRT-III filaments facilitate membrane remodelling for budding away from the cytosol and for repair of membrane damage. An early step in this process is the assembly of planar ESCRT-III spirals on the membrane surface. Low and colleagues have demonstrated that the bacterial Vipp1 protein, which is related to eukaryotic ESCRT-III subunits, form spirals on supported lipid bilayers (SLBs).

The remarkable work presented is a combination of light microscopy, fast-atomic force microscopy (F-AFM) and electron microscopy (EM). This combination is a powerful protocol for understanding how these proteins function dynamically on multiple scales. Previous work on Vipp1 structures showed the assembly of rings on membranes. The work of the present manuscript shows how these rings act as a nucleus for formation of Vipp1 spirals, and the authors propose that these rings colonize damaged regions of membranes to nucleate assemblies that can repair membranes.

The authors have characterised cryo-EM helical reconstructions of the protein subunits assembled on lipid tubules arising from lipid monolayers. The cryo-EM and associated model building appears to be of a high standard, and they provide 3.7 Å resolution reconstructions that are sufficient for the conclusions presented in the text. These observations go beyond the previously reported Vipp1 rings, and they suggest the conformational transitions from rings to spirals, planar assemblies and lipid tubules. These structures show clear relationships with eukaryotic ESCRT-III assemblies. The cryo-EM helical reconstructions carried out on samples introduced onto lipid monolayers show a variety of lattices present in samples, and the structural work provides a glimpse into how planar sheets of Vipp1 can be formed by opening the helical assemblies. These transitions are central to the many roles that ESCRT-III family subunits have in sculpting and repairing membranes. The work describes the helical structures that are made up of twisting filament, and their results suggest a how Vipp1 may stabilise and repair membranes. Their reconstructions of the C-terminal domain deletion variants compared with the full-length protein show that the domain has the effect of regulating Vipp1 polymer stability and dynamism.

Minor points

1. p. 8. Line 9. There is the term "striped filamentous filaments." I think this needs to be rephrased. I do not know what a non-filamentous filament would be.
2. Legend for figure 7C "Map contoured between 2.8-3.5s except helix a5c which was not post-processed." Does this mean the density in the region corresponding to alpha 5 was not present in the post processed map? I am not sure what it means to post process a helix.
3. p. 11, bottom "Concurrently, the number of ESCRT-III like protofilaments was reduced to a 14-start right-handed helix." I do not understand the meaning here. The 21-start helix becomes a 19-start helix on going from Vipp1Delta-a6_L2 and Vipp1Delta-a6_L3 to Vipp1L1. Where does the 14-start come from?
4. p. 12 bottom. "No map potential was located for C-terminal helix a6." Would be more understandable if it were "No potential density in the map was located for helix alpha6."
5. p. 12 bottom. "The key difference between Vipp1F197K/L200K_L1 and Vipp1L1 maps was the presence of lipid bilayer in the central lumen." Should be "The key difference between Vipp1F197K/L200K_L1 and Vipp1L1 maps was the presence of lipid bilayer in the central lumen of Vipp1F197K/L200K_L1"
6. p. 13. 4 lines from the bottom "Vipp1HR ring" Should probably be Vipp1HS ring
 In the same place "the transition between the latter two achieved by a remarkable lattice rotation and twist (Figure 6A)", There is nothing in Figure 6A that pertains to a transition between helical and ring polymers. Figure 6A just shows the interaction of the helical polymer with a membrane tubes. The transition is suggested in Figure 6B.
7. Top of p. 14. "Collectively, our results suggest a general mechanism for how Vipp1 may stabilise and repair membrane."
 This sentence does not seem to be related to anything else in the paragraph. However, the exact same sentence is the topic sentence of a paragraph at the bottom of p. 15, where it is related to the rest of the paragraph. I think the material on p. 14 may be cut-and-paste damage and should be removed.
8. In the methods, the authors have not given suppliers and catalog numbers for any of the reagents that they describe. This would be particularly useful for the lipids and microspheres.
9. The authors have used two different methods to prepare SLBs. In the first they added lipid microspheres onto mica sheets to allow the lipid bilayers to spill onto the mica. In the second they fused LUVs on mica sheets. The first was used for FLM and the second for F-AFM. Is there a reason that two different techniques were used? It may be that different groups used performed the two studies, and they used the method that was more familiar, but it would be helpful if they could comment on this.

Reviewer #3:
 Remarks to the Author:

The authors report intriguing findings, using mainly atomic force microscopy and electron microscopy, that show the mechanism of Vipp1 ESCRT-III family protein in forming planar sheets and spirals on membranes. This work also provides a mechanism by which the Vipp1 protein forms spirals and organizes into rings that can bud membranes.

First, the authors show that Vipp1 has a preference for highly curved and perturbed membranes by observing the recruitment of Vipp1-Alexa488 to supported lipid bilayers (SLB) and visualizing by confocal microscopy. In a second set of experiments, primarily by AFM, the authors observe that Vipp1 forms planar filaments that curl anticlockwise to form spirals and rings. The height and growth rate of the filaments were determined to be 5.5 nm and 19.6 nm/s respectively. It is interesting that the spirals did not form polygons as they grew to occupy most of the space. Rings likely formed once the spirals reached their curvature limit when curling was impeded. Vipp1 in high salt buffer remarkably can scan SLBs and localize to regions with patches, rupture lines- damage-. In contrast, Vipp1 rings in low salt buffer were sites for nucleating new Vipp1 architectures at the edge of membranes, rather than binding lipid. The Vipp1 ultrastructures of the sheet and spiral forms were both observed from AFM images to have parallel stripes or ridges spaces 54 angstroms apart, suggesting that this organization of the ridges is a key building block in the polymer/ultrastructure assembly. The Vipp1Da61-219 mutant, truncation of the CTD, forms tightly packed polymers of the spirals and sheets, supporting a role for the CTD in regulating Vipp1 dynamics. In a third set of experiments, using cryo-EM, the authors describe the molecular details of ring assembly and filament constriction by investigating four different Vipp1 helical filaments- Vipp1L1 and Vipp1F197K/L200K_L1 (an interface 3 mutant), Vipp1Da6_L2 and Vipp1Da6_L3. After overcoming the challenges of helical reconstruction by cryo-EM for these filaments, the authors observed that Vipp1Da6_L2 helical filaments are akin to the Vipp1 rings in high salt buffer observed under AFM. By comparing these filament structures, a mechanism of filament constriction is determined and the Vipp1F197K/L200K_L1 helical filament is constricted and tubulates membranes. The Vipp1L1 filament is constricted with an inner diameter of ~10.5 nm.

In summary, this work reports a mechanism for how the planar filaments transition to rings, using Vipp1 as a model for ESCRT-III family proteins. The Vipp1 likely functions in membrane repair by (1) the Vipp1 spirals encircling a damaged region and (2) the Vipp1 ring budding membranes.

Minor concerns: none

This work was presented clearly and well done with supporting data from atomic force microscopy and electron microscopy. The structural (cryo-EM) data is good as seen in the PDB validation report, statistics and other supporting data. Additionally, the methods are clearly described. As is rarely the case, I have no concerns with the manuscript.

1. One suggestion is to include the PDB files of the assembled state of Vipp1 for at least one rung of the helix. Such a visualization, and deposition to the PDB, would be helpful for non-structural biologists who may find it challenging to generate a helical polymer from the helical symmetry parameters, the cryo-EM map and models.

We would like to thank the reviewers very much indeed for taking the time to critique our manuscript so thoroughly and constructively. We really appreciate it. The manuscript has benefited from your comments.

Note: we have moved Supplementary Figure 1 into the main manuscript as Figure 1 as we think the addition of the light microscopy experiments strengthens the main paper and better acknowledges the contribution of the Roux lab. Therefore, Figure and Supplementary Figure numbers have been adjusted accordingly in the manuscript to reflect this change. We have also adjusted figure numbering referenced by the reviewers below to the updated numbering to avoid confusion. In the manuscript we have also switched nomenclature for Supplementary Figures to Extended Data Figures in line with NSMB nomenclature.

REVIEWER COMMENTS

Reviewer #1 (Remarks to the Author):

This paper uses F-AFM (usually referred to as HSAFM) and cryoEM to explore the dynamic behaviour of Vipp1 structures on lipid bilayers in exquisite detail. The major finding of this work is that the structure of VIPP1 filaments is dynamic and changes to accommodate the curvature required to form spirals, as has been observed for its homolog ESCRT. They show that these filaments can form sheets, which branch off into spiral forming filaments, and that these phases can co-exist together. They show that the sheets are formed of merged filaments, due to their regular spacing, which matches the width of the filament. They probe deeper into the spiral formation, demonstrating that at the centre of the spiral a protruding ring forms, by pinching off the spiral, and that these rings, when purified alone, preferentially bind to defects in membranes.

The paper then uses high resolution AFM (taken at lower speeds) to observe the regular crystalline structure of the filaments, and use fourier analysis to determine the crystal packing of the lattice. They show that the lattice structure is identical across the higher order phases of spirals, rings and sheets. They probe this deeper by analysing these structures with CryoEM, showing striking similarity between the AFM and CryoEM ridge spacing (54Å). They show that these can slide back and forth, with hairpins at each ridge dictating the curvature, by allowing the filaments to come closer or further apart (41-61 Å). Beyond this point, extreme curvature is induced by broken filaments, "wedging" or through tilt, which then leads to ring formation, causing the ring to protrude. This very detailed structural study is well evidenced throughout, I have listed below a number of minor points below that it would be useful for the authors to address:

AFM images do not have z scales - this is important to allow interpretation of the changing heights across the different structures

Z scales have been added to all AFM figures which do not showcase the phase.

For Fig. 2F-I : add the N for the data in the caption

N has been added for Fig. 2F-I in the legend. N has also been added for all other AFM-related graphs.

For the data in Fig. 2F it would be good to have an explanation (either in the text or the methods) of how the filament growth speed was calculated

To calculate filament growth speed, the segmented line function in Fiji was used to measure the change in filament length between movie frames. Each data point is a velocity between time points derived from 11 measured filaments. These sentences have been added to Methods.

Is there a reason that the N for Fig. 2G (filament width) is much lower than the other measured parameters?

We were limited in the number of measurable straight filaments which were sufficiently isolated from any neighbouring filament. However, with N=13 and a low standard deviation (0.9 nm) we are confident that this is a fair and accurate representation of filament width.

The height profile in Fig. 3B is missing the x axis label.

Thank you for spotting. X axis label (In-plane distance) now added.

A legend is required for Fig. 3H to show which line (blue or red) corresponds to the HS or LS rings

Legend has been added and the Y axis label colour coded to match the data lines.

The AFM in Fig. 4 is said to be taken at a lower scan rate - is it appropriate to continue referring to it as F-AFM at this point, or is it now just standard tapping mode AFM?

Fig. 4A was taken at 10Hz which is between Fast-AFM and conventional Tapping mode. Fig 4B was taken at 35Hz which is Fast-AFM. To distinguish how these images were obtained, we have added the refresh rate in Hz to the figure legend.

In Fig. 4B the red dotted boxes are difficult to see over the AFM images - you may want to consider using a different colour for these.

For colour consistency across the manuscript, we would like to stick with the red dotted box. However, we have increased the font width to 1.5 for all the red boxes on the figure so they are further highlighted. If this remains problematic for the reviewer please let us know and we can search for another solution.

It would be useful to make Fig. 5A larger, as currently it is hard to see the structures that are being highlighted. The same goes for Fig.6A

We have now enlarged Fig. 5A and 6A as much as possible in its current position. However, we acknowledge that this may still be insufficient to really see the detail. Having played with

various formats, we concluded that the optimal way to present the EM image and to showcase the different polymers was to present Fig. 5A and 6A in much larger format as Supplementary Figure 1A and 1B, respectively (not to be confused with Extended Data Figures). Supplementary Figure 1A and 1B have been referenced in the main text where appropriate. For consistency we have also included Extended Data Fig. 6A showing Vipp1_{L1} cryo image zoomed as Supplementary Figure 1C, also referenced in the main text.

In the AFM image in Fig. 7B, the numbers in red are not easily visible - consider changing the colour of this.

We have added white tabs/background to the red numbers in Fig. 7B left hand panel so that they should be readily visible now. We stuck with red as we think that the numbers 1-3 in the right hand panels, which relate to the numbers in the left hand panel, are best visible in this colour. Ultimately we needed a colour which stands out amongst a yellow panel (left) and panels with blue (right).

It would be important for the authors to add more detail to the AFM methods section. Given that high speed AFM was important to capture the dynamics of the Vipp, it would be important to provide the scan rate used. This is also true for the high resolution imaging carried out for Fig. 4. Additionally, the resolution that the AFM was taken at should also be provided.

The scan rate and pixel size for all AFM images have now been included in the figure legends.

Reviewer #2 (Remarks to the Author): Eukaryotic ESCRT-III filaments facilitate membrane remodelling for budding away from the cytosol and for repair of membrane damage. An early step in this process is the assembly of planar ESCRT-III spirals on the membrane surface. Low and colleagues have demonstrated that the bacterial Vipp1 protein, which is related to eukaryotic ESCRT-III subunits, form spirals on supported lipid bilayers (SLBs).

The remarkable work presented is a combination of light microscopy, fast-atomic force microscopy (F-AFM) and electron microscopy (EM). This combination is a powerful protocol for understanding how these proteins function dynamically on multiple scales. Previous work on Vipp1 structures showed the assembly of rings on membranes. The work of the present manuscript shows how these rings act as a nucleus for formation of Vipp1 spirals, and the authors propose that these rings colonize damaged regions of membranes to nucleate assemblies that can repair membranes.

The authors have characterised cryo-EM helical reconstructions of the protein subunits assembled on lipid tubules arising from lipid monolayers. The cryo-EM and associated model building appears to be of a high standard, and they provide 3.7 Å resolution reconstructions that are sufficient for the conclusions presented in the text. These observations go beyond the previously reported Vipp1 rings, and they suggest the conformational transitions from rings to spirals, planar assemblies and lipid tubules. These structures show clear relationships with eukaryotic ESCRT-III assemblies. The cryo-EM helical reconstructions carried out on samples introduced onto lipid monolayers show a variety of lattices present in samples, and the structural work provides a glimpse into how planar sheets of Vipp1 can be formed by opening the helical assemblies. These transitions are central to the many roles that ESCRT-III family subunits have in sculpting and repairing membranes. The work describes the helical structures that are made up of twisting filament, and their results suggest a how Vipp1 may stabilise and repair membranes. Their reconstructions of the C-terminal domain deletion variants compared with the full-length protein show that the domain has the effect of regulating Vipp1 polymer stability and dynamism.

Minor points

1. p. 8. Line 9. There is the term “striped filamentous filaments.” I think this needs to be rephrased. I do not know what a non-filamentous filament would be.

Thank you for spotting. We have removed filamentous.

2. Legend for Supplementary Figure 6C “Map contoured between 2.8-3.5s except helix a5c which was not post-processed.” Does this mean the density in the region corresponding to alpha 5 was not present in the post processed map? I am not sure what it means to post process a helix.

Due to high flexibility in Vipp1_{L1} helix a5c the map relating to this region is relatively low resolution/disordered and is degraded by sharpening or post processing with DeepEMhancer. For presentation purposes in the figure, we therefore show the map region relating to helix a5c without sharpening or post processing.

We agree the description was not clear. We have revised the Figure legend in Supplementary Figure 6 to: **C**, Vipp1_{L1} map fitted with Vipp1_{L1} helical filament structure (top left). The coloured monomer is isolated and zoomed to show map quality, build and fit. Map contoured between 2.8-3.5 σ . The map region relating to helix α 5c, contoured at 3 σ , was not DeepEMhancer processed or sharpened.

3. p. 11, bottom “Concurrently, the number of ESCRT-III like protofilaments was reduced to a 14-start right-handed helix.” I do not understand the meaning here. The 21-start helix becomes a 19-start helix on going from Vipp1Delta-a6_L2 and Vipp1Deltaa6_L3 to Vipp1L1. Where does the 14-start come from?

The sentence would have been clearer if written: “Concurrently, the number of ESCRT-III like protofilaments was reduced from a 17-start right-handed helix to a 14-start right-handed helix.” These Bessel Orders can be observed in Supplementary Figure 4 comparing left and right panels. However, as this is quite a technical detail and is not essential, we have removed the sentence to simplify the text. It is sufficient for readers to understand Vipp1_{L1} filament constriction occurring by the 21-start helix reducing to a 19-start helix (Extended Data Fig. 4).

4. p. 12 bottom. “No map potential was located for C-terminal helix α 6.” Would be more understandable if it were “No potential density in the map was located for helix α 6.”

Apologies for the confusion here. This relates to semantics between Coulomb potential EM maps vs electron density X-ray crystallography maps. We have removed the word potential so that the sentence now simply reads: No supporting map was located for C-terminal helix α 6.

5. p. 12 bottom. “The key difference between Vipp1F197K/L200K_L1 and Vipp1L1 maps was the presence of lipid bilayer in the central lumen.” Should be “The key difference between Vipp1F197K/L200K_L1 and Vipp1L1 maps was the presence of lipid bilayer in the central lumen of Vipp1F197K/L200K_L1”

Agreed and implemented.

6. p. 13. 4 lines from the bottom “Vipp1HR ring” Should probably be Vipp1HS ring

Agreed and implemented.

In the same place “the transition between the latter two achieved by a remarkable lattice rotation and twist (Figure 7A)”, There is nothing in Figure 7A that pertains to a transition between helical and ring polymers. Figure 7A just shows the interaction of the helical polymer with a membrane tubes. The transition is suggested in Figure 7B.

Agreed. We have cut reference to the transition here. The sentences now read:

“In addition, we show the structure of four different Vipp1 helical filaments with three related but different lattice assemblies. They show the close connection between 2D planar, helical and Vipp1_{HS} ring polymers. Given these different polymer forms are highly ordered, their comparison provides a basis for modelling how Vipp1 builds planar filaments that transition to 3D membrane budding forms through filament twist (Fig. 7A).”

7. Top of p. 14. “Collectively, our results suggest a general mechanism for how Vipp1 may stabilise and repair membrane.”

This sentence does not seem to be related to anything else in the paragraph. However, the exact same sentence is the topic sentence of a paragraph at the bottom of p. 15, where it is related to the rest of the paragraph. I think the material on p. 14 may be cut-and-paste damage and should be removed.

Agreed and removed.

8. In the methods, the authors have not given suppliers and catalog numbers for any of the reagents that they describe. This would be particularly useful for the lipids and microspheres.

Cat numbers have been included in the methods for lipids and microspheres.

9. The authors have used two different methods to prepare SLBs. In the first they added lipid microspheres onto mica sheets to allow the lipid bilayers to spill onto the mica. In the second they fused LUVs on mica sheets. The first was used for FLM and the second for F-AFM. Is there a reason that two different techniques were used? It may be that different groups used performed the two studies, and they used the method that was more familiar, but it would be helpful if they could comment on this.

In the FLM experiments, we aimed for a dynamic system, seeking to observe the adaptation of Vipp1 to growing lipids. However, in the case of Fast-AFM, using this system wasn't feasible due to the potential perturbation of AFM cantilevers by beads, leading to their breakage. Hence, we opted for LUVs to achieve the lipid bilayer in Fast-AFM experiments.

Reviewer #3:

Remarks to the Author:

The authors report intriguing findings, using mainly atomic force microscopy and electron microscopy, that show the mechanism of Vipp1 ESCRT-III family protein in forming planar sheets and spirals on membranes. This work also provides a mechanism by which the Vipp1 protein forms spirals and organizes into rings that can bud membranes.

First, the authors show that Vipp1 has a preference for highly curved and perturbed membranes by observing the recruitment of Vipp1-Alexa488 to supported lipid bilayers (SLB) and visualizing by confocal microscopy. In a second set of experiments, primarily by AFM, the authors observe that Vipp1 forms planar filaments that curl anticlockwise to form spirals and rings. The height and growth rate of the filaments were determined to be 5.5 nm and 19.6 nm/s respectively. It is interesting that the spirals did not form polygons as they grew to occupy most of the space. Rings likely formed once the spirals reached their curvature limit when curling was impeded. Vipp1 in high salt buffer remarkably can scan SLBs and localize to regions with patches, rupture lines- damage-. In contrast, Vipp1 rings in low salt buffer were sites for nucleating new Vipp1 architectures at the edge of membranes, rather than binding lipid. The Vipp1 ultrastructures of the sheet and spiral forms were both observed from AFM images to have parallel stripes or ridges spaces 54 angstroms apart, suggesting that this organization of the ridges is a key building block in the polymer/ultrastructure assembly. The Vipp1Da61-219 mutant, truncation of the CTD, forms tightly packed polymers of the spirals and sheets, supporting a role for the CTD in regulating Vipp1 dynamics. In a third set of experiments, using cryo-EM, the authors describe the molecular details of ring assembly and filament constriction by investigating four different Vipp1 helical filaments- Vipp1L1 and Vipp1F197K/L200K_L1 (an interface 3 mutant), Vipp1Da6_L2 and Vipp1Da6_L3. After overcoming the challenges of helical reconstruction by cryo-EM for these filaments, the overserved that Vipp1Da6_L2 helical filaments are akin to the Vipp1 rings in high salt buffer observed under AFM. By comparing these filament structures, a mechanism of filament constriction is determined and the Vipp1F197K/L200K_L1 helical filament is constricted and tubulates membranes. The Vipp1L1 filament is constricted with an inner diameter of ~10.5 nm.

In summary, this work reports a mechanism for how the planar filaments transition to rings, using Vipp1 as a model for ESCRT-III family proteins. The Vipp1 likely functions in membrane repair by (1) the Vipp1 spirals encircling a damaged region and (2) the Vipp1 ring budding membranes.

Minor concerns: none

This work was presented clearly and well done with supporting data from atomic force microscopy and electron microscopy. The structural (cryo-EM) data is good as seen in the PDB validation report, statistics and other supporting data. Additionally, the methods are clearly described. As is rarely the case, I have no concerns with the manuscript.

1. One suggestion is to include the PDB files of the assembled state of Vipp1 for at least one rung of the helix. Such a visualization, and deposition to the PDB, would be helpful for non-structural biologists who may find it challenging to generate a helical polymer from the helical symmetry parameters, the cryo-EM map and models.

We have deposited the asymmetric unit in the pdb database for the different filaments. However, we agree with the reviewer's concern. Therefore, for each atomic coordinate file we have included matrices for the biological assembly. Generating the helical filament is easy - in Chimera use the command `sym #N` or in ChimeraX⁷⁵ use `sym #N biomt` where N = model number. To help the reader we have now added instructions to the Data and code availability section:

Atomic coordinates have been deposited in the Protein Data Bank (PDB) under accession code 8QBR, 8QBS, 8QBV and 8QBW, respectively. Note that each atomic coordinate file includes matrices for the biological assembly. To generate the helical filament in Chimera use the command `sym #N` or in ChimeraX⁷⁵ use `sym #N biomt` where N = model number.

Decision Letter, first revision:

Message: Our ref: NSMB-A48327A

5th Mar 2024

Dear Dr. Low,

Thank you for submitting your revised manuscript "Mechanism for Vipp1 spiral formation, ring biogenesis and membrane repair" (NSMB-A48327A). It has now been seen by the original referees and their comments are below. The reviewers find that the paper has improved in revision, and therefore we'll be happy in principle to publish it in Nature Structural & Molecular Biology, pending minor revisions to satisfy the referees' final requests and to comply with our editorial and formatting guidelines.

[EDITOR: REMOVE IF WORD DOCUMENT IS AVAILABLE]

To facilitate our work at this stage, it is important that we have a copy of the main text as a word file. If you could please send along a word version of this file as soon as possible, we would greatly appreciate it; please make sure to copy the NSMB account (cc'ed above).

Sincerely,

Katarzyna Ciazynska, PhD
(she/her)
Associate Editor
Nature Structural & Molecular Biology
<https://orcid.org/0000-0002-9899-2428>

Reviewer #1 (Remarks to the Author):

The authors have addressed all of our concerns and we are happy to recommend this paper for publication

Dr Alice Pyne and Dr Thomas Catley

Reviewer #2 (Remarks to the Author):

The authors have addressed all of my concerns, and responded to all of the issues raised by the other reviewers.

Reviewer #3 (Remarks to the Author):

The authors have addressed my minor concern and provided a means for generating the biological assemblies for each atomic coordinate file.

Final Decision Letter:

Message: 11th Sep 2024

Dear Dr. Low,

We are now happy to accept your revised paper "Mechanism for Vipp1 spiral formation, ring biogenesis and membrane repair" for publication as an Article in Nature Structural & Molecular Biology.

Due to the importance of these deadlines, we ask that you please let us know now whether you will be difficult to contact over the next month. If this is the case, we ask you provide us with the contact information (email, phone and fax) of someone who will be able to check

the proofs on your behalf, and who will be available to address any last-minute problems.

Your paper will be published online soon after we receive proof corrections and will appear in print in the next available issue. You can find out your date of online publication by contacting the production team shortly after sending your proof corrections.

If you have not already done so, we strongly recommend that you upload the step-by-step protocols used in this manuscript to the Protocol Exchange. Protocol Exchange is an open online resource that allows researchers to share their detailed experimental know-how. All uploaded protocols are made freely available, assigned DOIs for ease of citation and fully searchable through nature.com. Protocols can be linked to any publications in which they are used and will be linked to from your article. You can also establish a dedicated page to collect all your lab Protocols. By uploading your Protocols to Protocol Exchange, you are enabling

researchers to more readily reproduce or adapt the methodology you use, as well as increasing the visibility of your protocols and papers. Upload your Protocols at www.nature.com/protocolexchange/. Further information can be found at www.nature.com/protocolexchange/about.

Please note that *Nature Structural & Molecular Biology* is a Transformative Journal (TJ). Authors may publish their research with us through the traditional subscription access route or make their paper immediately open access through payment of an article-processing charge (APC). Authors will not be required to make a final decision about access to their article until it has been accepted. Find out more about Transformative Journals

Sincerely,

Kat

Katarzyna Ciazynska, PhD
(she/her)
Senior Editor
Nature Structural & Molecular Biology
<https://orcid.org/0000-0002-9899-2428>